# KBFormer: A Transformer-based Diffusion Model of Structured Entities with Heterogeneous Properties

## Abstract

We develop a generative attention-based approach to modeling structured entities comprising different property types, such as numerical, categorical, string, and composite. This approach handles such heterogeneous data through a mixed continuous-discrete diffusion process over the properties. Our flexible framework is capable of modeling entities with arbitrary hierarchical properties, enabling applications to structured Knowledge Base (KB) entities and tabular data. Our approach obtains state-of-the-art performance on a majority of cases across 15 datasets. In addition, experiments with a device KB and a nuclear physics dataset demonstrate the model's ability to learn representations useful for entity completion in diverse settings. This has many downstream use cases, including modeling numerical properties with high accuracy - critical for science applications, which also benefit from the model's inherent probabilistic nature.

## 1 Introduction

Deep generative models refer to a family of generative machine learning (Ng & Jordan, 2001) approaches that learn a joint distribution over the input space using deep neural networks (Goodfellow et al., 2016). Examples of these models include large language models (LLMs) for text (OpenAI, 2023; Thoppilan et al., 2022; Chowdhery et al., 2022; Touvron et al., 2023a), as well as generative models for other modalities, such as for vision (Rombach et al., 2022; Ramesh et al., 2022) and audio (Oord et al., 2016; Kim et al., 2018; Ping et al., 2020). In this work, we explore generative modeling of structured entities with heterogeneous properties, such as entries in rich knowledge bases (KBs), items in product catalogs, or scientific catalogs, and ontologies like the periodic table of elements and the various properties of isotopes. A structured entity in this context contains—and is represented by—a set of associated properties, where each property has a key that belongs to a pre-defined global schema and a value. The schema associates a specific datatype—*e.g.*, string, categorical, numerical, or a composite of other datatypes—with each property key that corresponding property values must adhere to.

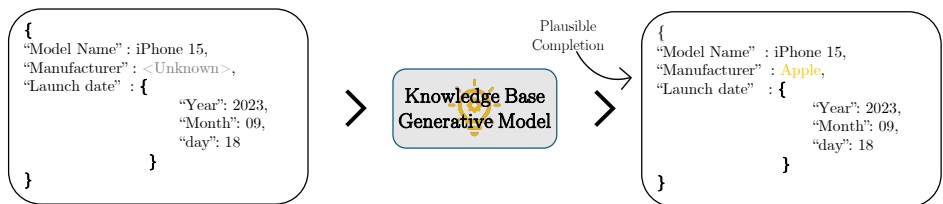

Figure 1: Masked entity modeling using a generative model to get plausible property completions.

A model of joint distribution over properties may have many applications. Such a model can, for example, be used to automatically infer missing property values for entities in a KB, often referred to as the KB completion task, as shown in Figure 1. It may also be employed during KB construction to predict if two different entity fragments correspond to the same real-world entity—*i.e.*,

for entity linking. A model of joint distribution over properties may also be useful for detecting anomalous property values in the dataset. Finally, these generative models also learn useful latent representations of properties and entities that other foundation models (Bommasani et al., 2021) could consume, such as an LLM in a KB-augmented question-answering task. In this work, we propose a Transformer-based architecture that allows us to cross-attend over the hierarchical structure of each entity and includes specific encoder-decoder models appropriate for each basic datatype. To summarize, the key contributions of our work are:

- We propose a hybrid diffusion training paradigm that allows for joint modeling of an entity's properties in a principled manner.
- We develop a framework for handling heterogeneous property types along with semantic hierarchical encodings for different property types.
- We employ an attention-based architecture, dubbed *KBFormer*, to enable the sharing of information across properties and demonstrate various downstream applications of the model, such as high-precision predictions and generating high-quality synthetic samples from tabular data.

## 2 RELATED WORK

There is prior work on generative modeling of tabular data (Kotelnikov et al., 2023; Lee et al., 2023; Xu et al., 2019) that shares several characteristics with generative modeling of structured entities, insofar as a row in the tabular data can be considered as an entity and the corresponding heterogeneous column values as its set of properties. However, in structured entities, a property may also be composed of other datatypes—*e.g.*, a quantity is a composite of a numerical value and a unit of measurement (categorical type), and a date may be represented as a composite of three numerical values,[1] *i.e.*, day, month, and year—which implies a richer hierarchical structure compared to a row in a typical tabular dataset. See Figure 2 for an illustration. Previous work on generative models for tabular data has largely focused on the scenario where each row is a fixed-size vector of only numerical and categorical values that can be flattened into a simple feature vector. Unlike these previous works, we are interested in modeling richer datatypes, including text- and composite-datatypes. The framework we propose is flexible and extensible and, though outside the scope of this paper, can be used for large-scale pre-training on large and varied collections of KB.

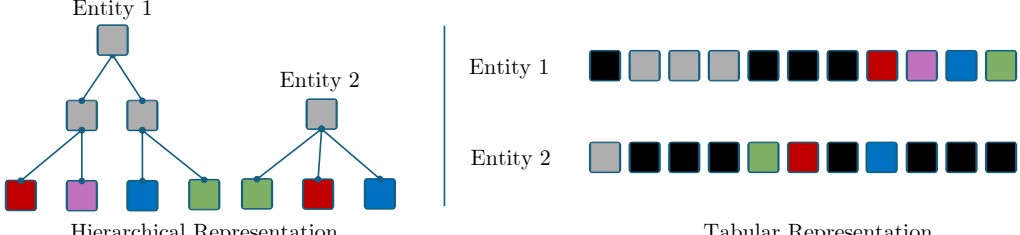

Figure 2: Hierarchical representations of entities can model rich relationships that can be difficult to capture with dense tabular representations, which can be prohibitive for sparse KBs. Black squares correspond to non-existing values, making the table sparse.

Another relevant line of work is Knowledge Base (KB) modeling which is often viewed as a link prediction problem, where the knowledge base is represented as a collection of factual triples (head, relation, tail) (Bordes et al., 2013; Lin et al., 2015; Sun et al., 2018; Schlichtkrull et al., 2018; Nathani et al., 2019). This often necessitates a high-quality subset of triples, and conventional models may struggle to generalize to generating facts with entirely new entity tails. In this work, we take a more direct approach and model the knowledge base as a collection of entities, framing knowledge modeling as a masked property prediction task over incomplete entity representations. The proposed training procedure and model architecture are designed to capture interdependence between properties. Furthermore, our new evaluation scheme focuses on capabilities relevant to knowledge-intensive tasks. For instance, instead of deciding whether links are factual or not, this approach focuses on entity completion from prior associations. Our KB model does not simply learn to assess the validity of a particular triplet; it learns algorithms to derive an entity's properties.

---

[1]It may also make sense to represent day and month as categorical types.

## 3 GENERATIVE MODELING OF STRUCTURED ENTITIES

A common approach in the literature for generating new facts is to evaluate the validity of triples using link prediction models. Instead, we take a generative perspective on knowledge base modeling. In this section, we describe the training procedure of a generative model with (fixed-mask) masked modeling. We then show a simple loss modification to formulate the problem as an absorbing continuous-time diffusion over discrete states. This formulation allows samples to be more consistent and of higher quality by smoothing the generative process over small steps in the number of properties unmasked per iteration.

### 3.1 MASKED MODELING

A naive approach to training our entity completion model, parameterized by $\theta$, is to directly predict hidden properties based on a subset of available properties using a fixed mask, similar to Devlin et al. (2019). Consider an entity $\mathbf{x} \sim p(x)$ where each dimension corresponds to a property. At each training step, the model is given a collection of properties associated with the entity, $\tilde{\mathbf{x}}$. Some property values are replaced with a special mask token. The model then predicts the true values of the masked properties conditioned on the visible properties

$$\mathcal{L}_{CT} = \mathbb{E}\left[ \sum_{d|\tilde{x}^d \text{ is masked}} -\log p^\theta\left(x^d \mid \tilde{\mathbf{x}}\right) \right],\tag{1}$$

which amounts to a per-property reconstruction loss (*e.g.*, using cross-entropy or mean squared error). At inference time, the model is used exactly in the same fashion. A collection of properties is given and the model predicts all remaining properties in a single step.

This approach is not necessarily optimal in terms of the quality of generated samples. Single-step models can be inferior to traditional left-to-right autoregressive models, so it is natural to expect improved quality of generated samples if the model is allowed to fill in the missing properties autoregressively (Ghazvininejad et al., 2019). At the cost of more computation, the diffusion approach we will formalize in the next section solves this issue (see Appendices B.1.2, for an intuitive example, and B.1.1 for a full ablation).

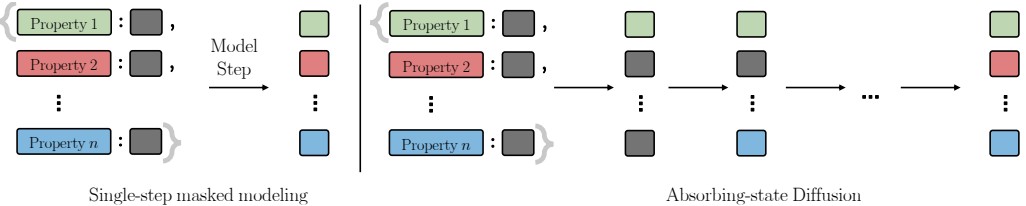

Single-step masked modeling                    Absorbing-state Diffusion

Figure 3: Generating samples with masked modeling in one step (left) and autoregressively (right). Note that properties are unmasked in random order.

### 3.2 A FORMULATION OF DIFFUSION OVER HETEROGENOUS DATA

We now explore a diffusion modeling approach based on Discrete Denoising Diffusion Probabilistic Models (D3PM) (Austin et al., 2021), specifically continuous-time discrete-state diffusion (Campbell et al., 2022) with an absorbing state. In the following, we will define a diffusion paradigm that can be seen as an autoregressive extension of the masked modeling approach. In Figure 3, we see how both setups can be used to sample new entities.

Our proposed diffusion process can be summarized as **(1)** For an entity $\mathbf{x}_0 \sim p(\mathbf{x}_0)$ with $D$ dimensions, individual properties randomly flow into the absorbing masked state. At the end of this process, all properties are masked. **(2)** The reverse process, which is completely defined by the forward process but is generally intractable, is modeled via a parameterized conditional distribution, at step $t$, $p^\theta(\mathbf{x}_{t-1}|\mathbf{x}_t)$ as the random de-masking of individual properties, $x^d$. The objective is to maximize the log-likelihood of the data under this reverse conditional.

**Forward Process**  The noising process randomly masks an entity's properties at a sampled rate. Surprisingly, the objective reduces to the standard reconstruction loss weighted by masking amount. The complete derivation is available in Appendix A.2; this section outlines the initial steps. Consider the forward process $q(\mathbf{x}_t|\mathbf{x}_{t-1})$ from $p(\mathbf{x}_0)$, the data distribution, towards an easy-to-sample reference distribution $q(\mathbf{x})$ with all dimensions (properties) in the masked state. We apply a continuous-time diffusion to dimension $d$ of $\mathbf{x}_0$ with a transition rate matrix

$$R_t^d = \begin{bmatrix} 0 & 0 & 0 \\ -\beta(t) & \beta(t) & 0 \\ -\beta(t) & 0 & \beta(t) \end{bmatrix},$$

where, for illustration purposes, we used a discrete distribution with 3 states and reserved the state 0 for the mask state. Each state flows into the absorbing mask state with the same rate $\beta(t)$. Solving Kolmogorov's equations, reveals the marginal distribution for the state at time $t$ conditioned on the initial state is given by $\mathbf{x}_0^T P_t$ (using a slight abuse of notation to handle all dimensions simultaneously) where

$$P_t^d = \exp \int_0^t R_s^d ds = \begin{bmatrix} 1 & 0 & 0 \\ 1-e^{-\gamma(t)} & e^{-\gamma(t)} & 0 \\ 1-e^{-\gamma(t)} & 0 & e^{-\gamma(t)} \end{bmatrix}, \qquad (2)$$

with $\gamma(t) = \int_0^t \beta(s) ds$. At time $t$, property $d$ jumps into a masking state with probability $1-e^{-\gamma(t)}$, while masked properties remain absorbed. We set $\beta(t)$ such that at time $t = 1$, all properties are masked (i.e., $\gamma(1) = \infty$). The specific $\beta(t)$ is irrelevant as we can integrate across the masking probability instead of time, similar to integrating across the signal-to-noise ratio (SNR) in Kingma et al. (2021) for Gaussian diffusion models. Though the forward process is independent for each dimension, it is useful to denote the joint rate over properties

$$R_t(\mathbf{x}, \tilde{\mathbf{x}}) = \sum_{d=1}^{D} \delta(\mathbf{x}^{\neg d}, \tilde{\mathbf{x}}^{\neg d}) R_t^d(x^d, \tilde{x}^d), \qquad (3)$$

where $\neg d$ indexes is a vector of all properties from 1 to $D$ except property $d$. Here, the Kronecker delta $\delta$ is used to specify that there is no change from one vector to another unless exactly one property, $d$, changes, in which case the rate is given by the rate matrix for that property. The total rate of change across all properties is then given by

$$Z_t(\mathbf{x}) = \sum_{\mathbf{x} \neq \tilde{\mathbf{x}}} R_t(\mathbf{x}, \tilde{\mathbf{x}}) = \beta(t)(D - N_t), \qquad (4)$$

where $N_t$ is the number of masked properties at time $t$. It is also useful to write the probability of transitioning from state $\mathbf{x}$ to $\tilde{\mathbf{x}}$ at time $t$ as $r_t(\tilde{\mathbf{x}}|\mathbf{x}) = (1 - \delta(\mathbf{x}, \tilde{\mathbf{x}}))R_t(\mathbf{x}, \tilde{\mathbf{x}})/Z_t(\mathbf{x})$. We can also define the empirical masking rate $\hat{\pi} = N_t/D$. These expressions will be useful in deriving the likelihood bound in Proposition 1 (see Appendix A for more details).

**A simple simulation of the backward diffusion process**  With our forward process defined, we focus on the backward process. We know from our choice of the forward process that at time $t = 1$ the state will be all masks with probability 1. In the reverse process, Equation 8 (see Appendix A.1) tells us that once a property has been de-masked, it will stay de-masked until $t = 0$. Masked properties transition to de-masked states at a rate proportional to the model's prediction given the current state. Because all the properties flow at the same rate, the order in which the properties are de-masked is random, irrespective of the model. As we approach $t = 0$, the rate approaches infinity, fully de-masking all properties by $t = 0$.

A simple algorithm can implement the reverse diffusion process as follows: First, initialize with a sequence comprising entirely masked states. Then, randomly select a masked property and predict its new state using the neural network, conditioned on the current states of all properties. Replace the selected property's masked state with its newly predicted state. Repeat this process, picking randomly masked properties and predicting their unmasked states until no masked states remain. While this simulation disregards event timing, that omission is inconsequential for our purposes. Unmasking is not restricted to removing one mask at a time; instead, we can employ multiple leaps ($> 1$) in every step (Campbell et al., 2022). When the leap count is equal to the number of properties, the reverse diffusion process is equivalent to the generative step from the masked modeling procedure, and we simply predict all properties at once. Although this approach may offer computational advantages, it could also weaken the correlations that maintain the samples' consistency (see Figures 6 and 7 as well as the ablation study in Appendix B.1).

**Likelihood bound** The choice of absorbing state kernel yields a surprisingly simple likelihood bound, which can be written as a denoising loss weighted by the amount of masking noise.

**Proposition 1** *For the reverse diffusion from the fully masked stationary distribution towards $p(\mathbf{x}_0)$, an upper bound on the model negative loglikelihood $\mathbb{E}_{p(x)}[-\log p_0^\theta(x)]$ can be given by*

$$\mathcal{L}_{CT} = \mathbb{E}_{\pi \sim \mathcal{U}(0,1),\, \tilde{\mathbf{x}} \sim \psi(\tilde{\mathbf{x}})} \left[ D \frac{1 - \hat{\pi}}{1 - \pi} \frac{1}{N_t + 1} \sum_{d | \tilde{x}^d = 0} - \log p_{0|t}^\theta \left( x_0^d \mid \tilde{\mathbf{x}} \right) \right], \quad (5)$$

*where $\psi(\tilde{\mathbf{x}}) = \sum_{\mathbf{x}} q_t(\mathbf{x}) r_t(\tilde{\mathbf{x}}|\mathbf{x})$.*

A full proof is available in Appendix A. The terms in green (under the expectation) are a direct implementation of the simulation process described in detail in Appendix A.2, the term in blue (the prefactor to the sum) is a simple rescaling factor, and the term in red (the sum) is the usual reconstruction loss but with a random masking rate.

**A Continuous Relaxation of Discrete State Diffusion** So far, we have only discussed discrete-state diffusion, valid for categorical properties. Here, we turn our attention to numerical properties. To predict numerical values to a high degree of precision, we can choose a discretization with a large but finite number of bins and employ all the machinery we developed up to this point. The full softmax can become quite expensive to evaluate in this case. Though there are several ways to alleviate this issue, such as hierarchical softmax (Morin & Bengio, 2005) or various contrastive alternatives (Oord et al., 2018; Sohn, 2016; Oh Song et al., 2016; Schroff et al., 2015), we will instead approximate the softmax when the number of bins (classes) tends to infinity.

We will end up using a Gaussian Mixture Model (GMM) for numerical properties. But first, to develop some intuition, we will explore a simplified approach where we assume we only want to model Gaussian numerical properties. A reasonable categorical model of continuous values captures ordinal properties and approaches Gaussian uncertainty in the limit of a large number of classes. Suppose the "correct" target value is $x$, we can take the discrete distribution $P(b_i) \propto \exp -||x - b_i||^2$, where $b_i$ is the bin-center of the $i$-th bin. In this case, we can write out the cross-entropy loss over bins, assuming $x$ is in the $i$-th bin, as follows $-\log P(b_i) = -\log \text{softmax}(-(\mathbf{b} - x\mathbf{1})^2)_i$, which is simply $-\log \exp(-(b_i - x)^2)$, ignoring the normalization, we can use squared error $(b_i - x)^2$ as a de-masking loss. In this setup, optimizing the cross-entropy over a large number of bins amounts to optimizing the center of the target bin using the mean-squared-error (MSE), avoiding a potentially prohibitive computation. With our loss function and generative procedure defined, we can move our focus to the neural architecture we will employ.

## 4 KBFORMER

This section describes the KBFormer model architecture, with Figure 4 showing a high-level overview. KBFormer takes an entity with an arbitrary number of properties of any type (any of which can be missing or "masked"). The property keys are used to generate semantic hierarchical encodings, and property values are passed to an encoder for the appropriate type. The outputs of both encoding steps are added together. For missing or masked properties, values are not encoded and only their hierarchical encodings are used. A transformer module aggregates information across properties and each element gets decoded into the appropriate probabilistic parameters we would later sample from *i.e.*, GMM means, variances, and weights for numerical properties and logits for text and categorical properties. Only masked properties are decoded, and the loss is evaluated on them. We will now provide more details on each step (see Appendix C for technical details).

**Hierarchical positional encoding** Our goal is for the model to understand entities' properties semantically. Once a schema is created, the hierarchical encodings are generated using a sequence model; in our setup, we use a simple RNN over the path to the node of interest (see Figures 4 and 9). Alternatively, language model representations could be leveraged, reading off the hierarchical encoding from a special token like BERT's [CLS] token (Devlin et al., 2019).

**Encoding** Each property value will first be embedded. Embedding schemes differ for each type of input, in this case, text, categorical and numerical. Categorical variables, just like text tokens, are

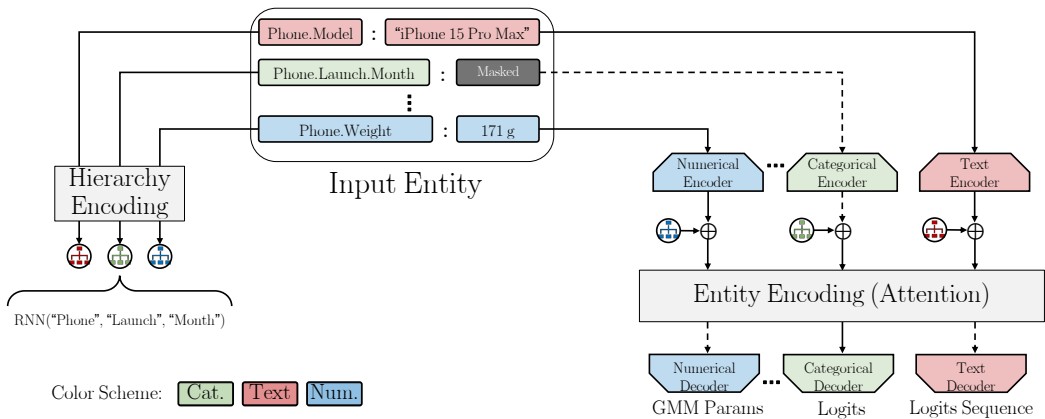

Figure 4: The KBFormer architecture. Keys from the input entity are used by an RNN to generate hierarchical encodings (left). They are then added to encoded values (right) the result is processed by an encoder and type-specific decoders output logits and GMM parameters. Dashed lines are not computed *i.e.,* masked values are not encoded and unmasked values are not predicted.

one-hot encoded, as is standard in language modeling. Numerical values need to be treated differently in order to preserve their numeracy properties. DICE (Sundararaman et al., 2020) embeddings or the standard sinusoidal embeddings (Vaswani et al., 2017) with learnable frequencies are good options. Though DICE embeddings preserve magnitude and order by construction, we could not find conclusive evidence for the superiority of either approach (see ablations in Appendix B.2). Each property value passes through an encoder module tailored to that property type. This can be achieved either via conditioning on the property itself (through the hierarchy positional encoding, for instance) or via disjoint encoders for each property. We opt for the latter in our experiments. The encoder architecture suits the input modality, mapping inputs to fixed-dimensional vector representations. We use MLPs with residual connections for categorical and numerical properties and we extract the first token representation from a Transformer encoder for text fields. Note that we can also use pre-trained language models as encoders for text properties but that is beyond our scope. These encoders map heterogeneous properties into a shared entity latent space. Their role is to transform arbitrary input types into a common representation format.

**Entity encoding and decoding to property values** After encoded properties have been augmented with positional encodings, multiple Transformer encoder layers process the properties. Masked elements are not attended to. With full entity context, property encodings are decoded via specialized decoders. Again we have the option of tying them (with conditioning), but we choose to use disjoint modules. Decoders output probabilistic parameters - logits for categorical/text, and GMM parameters ($\mu$, $\sigma$, weight) for numerical values. These parameters define the distributions we can sample from in the reverse process. For text properties, we use a transformer decoder to generate a sequence of tokens conditioned on the property encoding.

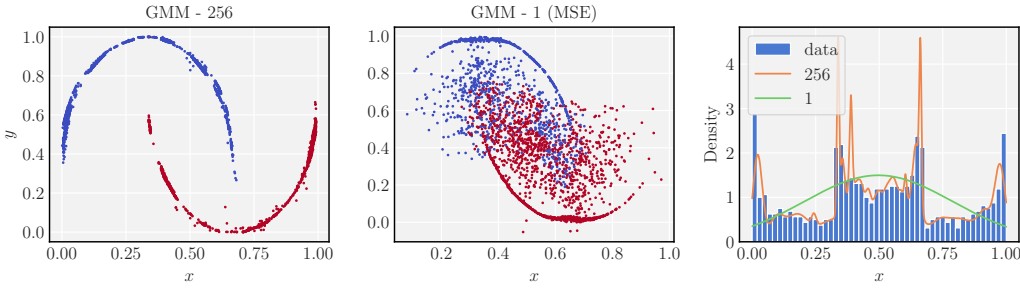

Figure 5: Generated samples using a KBFormer with GMM likelihood. The GMM uses 256 components (left) or 1 component (middle) which is equivalent to MSE when we fix the variance to unity. (right) A histogram of the data with the KBFormer learned marginals.

**Revisiting Numerical Properties**   In section 3.2, we made some choices to approximate the soft-max computation for numerical quantities with an MSE. However, the cost of this simplification is a model with low expressivity. In particular, the diffusion model we just constructed will be unable to capture a possible multi-modality of the marginal of some property $d$. This can be remedied by using a Gaussian Mixture Model (GMM) with more components. In Figure 5, we show a toy KBFormer model trained to generate samples from the two moons dataset. We can think of each point as an entity with the $x$ and $y$ coordinates treated as numerical properties and the class as a categorical property. Unlike the model trained with MSE, the model trained with a GMM likelihood can capture the multiple modes of the marginal of $x$.

## 5   EXPERIMENTS

**The Efficiency of Structure**   First, we show experiments using the Kaggle GSMArena dataset (Appendix F). We show that, when dealing with structured data (such as knowledge bases), the structured KBFormer architecture outperforms baselines consisting of unstructured text decoder-only models. We perform the experiment using both a fine-tuned LLaMA2-7B model (Touvron et al., 2023b) and a small decoder-only model trained from scratch. In particular, it is evident that at a lower parameter count, the structured inductive bias provides gains over the unstructured baseline at much larger scales. Furthermore, we show that we get favorable performance compared to Gradient-Boosted Decision Trees, which offer state-of-the-art performance on tabular data. The GBDTs are trained to predict a single property based on all others, offering a strong baseline. Unlike KBFormer, GBDTs do not handle missing data naturally, which can explain the performance gap (see  F for more training details). All models use a 80-20 train-test split. The decoder-only architectures (pre-trained and randomly initialized) use next-token prediction. LLaMA inputs are JSON-formatted string representations, with properties (key-value pairs) permuted 10 times before tokenization to augment the training set. Without this augmentation, causal models achieve worse performance because of difficulties in knowledge manipulation (Zhu & Li, 2023). To evaluate predictions on a particular property, we prompt the model with all other property keys and values followed by the key of the property we aim to predict. The model must output the property value and a closing brace to form valid JSON. See additional details in Appendix F.1. Note that this evaluation heavily favors the autoregressive models. These choices aim to present a challenging "best effort" scenario, establishing a difficult benchmark to surpass. Naturally, KBFormer is capable of making predictions using a much smaller number of properties. Figure 12 shows the metrics as a function of the proportion of properties given.

| Field | KBFormer | Decoder no pre-train | LLaMA2-7B | LLaMA2-7B 0-shot | GBDT Baseline |
|---|---|---|---|---|---|
| weight | $\mathbf{20.8}_{\pm \mathbf{0.5}}$ | 71.9 | 24.2 | 62.4 | $25_{\pm 1}$ |
| height | $\mathbf{5.7}_{\pm \mathbf{0.2}}$ | 79.6 | 6.4 | 94.4 | $6.2_{\pm 0.2}$ |
| depth | $1.67_{\pm 0.05}$ | 3.90 | 1.82 | 7.11 | $\mathbf{1.65}_{\pm \mathbf{0.04}}$ |
| width | $3.5_{\pm 0.3}$ | 42.7 | 4.110 | 69.6 | $\mathbf{3.8}_{\pm \mathbf{0.2}}$ |
| display-size | $\mathbf{0.64}_{\pm \mathbf{0.02}}$ | 7.47 | 0.707 | 10.5 | $1.08_{\pm 0.05}$ |
| battery | $\mathbf{0.233}_{\pm \mathbf{0.008}}$ | 6.99 | 0.257 | 969 | $0.304_{\pm 0.007}$ |
| launch.day | $11_{\pm 1}$ | 30.9 | 11.27 | 15.0 | $\mathbf{8.7}_{\pm \mathbf{0.2}}$ |
| launch.month | $3.4_{\pm 0.02}$ | 4.79 | 5.11 | 66.7 | $\mathbf{3.281}_{\pm \mathbf{0.005}}$ |
| launch.year | $\mathbf{1.25}_{\pm \mathbf{0.06}}$ | 947 | 1.22 | 458 | $1.42_{\pm 0.02}$ |
| oem | $\mathbf{0.181}_{\pm \mathbf{0.005}}$ | 0.484 | 0.231 | 0.711 | $0.4_{\pm 0.02}$ |
| network-edge | $0.221_{\pm 0.005}$ | 0.371 | $\mathbf{0.217}$ | 1.000 | $0.75_{\pm 0.01}$ |
| model | $\mathbf{0.881}_{\pm \mathbf{0.004}}$ | 0.900 | $\mathbf{0.878}$ | 0.928 | – |
| parsing error rate | $\mathbf{0}\%$ | 3.6% | 3.9% | 17.0% | $\mathbf{0}\%$ |
| num. parameters | 24.8M | 30.7M | 7B | 7B | – |

Table 1: Comparison of property prediction via causal decoder-transformer and structured generative approaches. Numerical properties (above line) use RMS error. Categoricals (below line) use error rate (1 - accuracy). "model" is a text field and uses word-based intersection over union (IoU) since tokenizers differ. To ensure that smaller values are better for all fields, we use 1 - IoU for text. Parsing error rate measures invalid JSON string predictions.

The unstructured decoder-only architectures demonstrate impressive prediction capabilities, especially LLaMA, though pretraining may enable data leakage. We provide a zero-shot LLaMA experiment to gauge this effect. However, those decoders lack critical inductive biases for structured data like the ordinality of numerical properties. In contrast, KBFormer incorporates these properties, improving performance on structured data versus LLaMA, as Table 1 shows. This highlights the importance of embedding structure-aware inductive biases, even at massive scale. While decoder-only models can memorize statistical regularities, their lack of inherent constraints results in exploding errors without augmentation. They also struggle to output valid, parseable entities, particularly at small scales. KBFormer's structured formulation prevents these issues by design.

**Nuclear Physics Predictions** Many scientific applications lack large-scale data due to the difficulty of taking measurements, the rarity of the events measured, or the prohibitive cost of obtaining more data. We will explore the benefits of using a KB generative model to learn from limited data. Nuclear properties are a good example, and developing accurate models for them can have a large impact on many subfields of physics, such as nuclear (astro)physics, including $r$-process nucleosynthesis Burbidge et al. (1957), the nuclear neutron skin and its consequences for the structure of neutron stars Brown (2000); Horowitz & Piekarewicz (2001); Gandolfi et al. (2012), the exploration of the boundaries of the nuclear landscape Erler et al. (2012), etc. Here we tackle the knowledge completion task, on a nuclear physics dataset, comprising 3254 nuclei. The features that we predict here are categorical and numerical in nature, detailed in Appendix E.

To our knowledge, no single model predicts the diverse physical properties we consider. However, specialized binding energy models provide reasonable baselines, with errors from 140

| Field | KBFormer | TDDPM | Optimal Const. | GBDT Baseline |
|---|---|---|---|---|
| $E_b$ [keV] | **370**$_{\pm 40}$ | $1700_{\pm 70}$ | 5570 | $640_{\pm 40}$ |
| Radius [fm] | **0.011**$_{\pm 0.001}$ | $0.445_{\pm 0.008}$ | 0.717 | $0.169_{\pm 0.009}$ |
| $t_{1/2}$ [logsec] | **1.51**$_{\pm 0.01}$ | $2.63_{\pm 0.02}$ | 3.63 | $1.72_{\pm 0.09}$ |
| Spin | $1.2_{\pm 0.1}$ | $1.78_{\pm 0.02}$ | 1.74 | **1.02**$_{\pm 0.01}$ |
| Abundance | **10.8**$_{\pm 0.1}$ | $13.7_{\pm 0.1}$ | 14.8 | **10**$_{\pm 1}$ |
| $Q_\alpha$ [keV] | **360**$_{\pm 50}$ | $1330_{\pm 30}$ | 6592 | $1290_{\pm 40}$ |
| $Q_{\beta^-}$ [keV] | **310**$_{\pm 20}$ | $2350_{\pm 80}$ | 7781 | $1790_{\pm 80}$ |
| $Q_{\beta^- + n}$ [keV] | **440**$_{\pm 80}$ | $2800_{\pm 200}$ | 10558 | $2300_{\pm 100}$ |
| $Q_{\mathrm{EC}}$ [keV] | **520**$_{\pm 40}$ | $2340_{\pm 80}$ | 7643 | $1900_{\pm 100}$ |
| $\beta_2$ [barns] | $0.93_{\pm 0.02}$ | $1.26_{\pm 0.02}$ | 1.36 | **0.43**$_{\pm 0.02}$ |
| Volume | **0.8**$_{\pm 0.1}$ | $3_{\pm 1}$ | 66.49 | **0.88**$_{\pm 0.05}$ |
| Surface | $0.21_{\pm 0.02}$ | $0.5_{\pm 0.1}$ | 8.763 | **0.127**$_{\pm 0.007}$ |
| Symmetry | **0.218**$_{\pm 0.002}$ | $0.28_{\pm 0.04}$ | 4.137 | $0.35_{\pm 0.02}$ |
| Coulomb | **5.3**$_{\pm 0.6}$ | $6_{\pm 1}$ | 482.8 | $11_{\pm 0.6}$ |
| Stability | $0.01_{\pm 0.001}$ | $0.088_{\pm 0.005}$ | 0.076 | **0.004**$_{\pm 0.001}$ |
| Parity | **0.047**$_{\pm 0.003}$ | $0.36_{\pm 0.01}$ | 0.68 | $0.077_{\pm 0.007}$ |

Table 2: Performance on the Nuclear Physics dataset. RMS values for numerical values above the line and errors for categorical features below. Properties without a unit specification have no units. Volume, Surface, Symmetry and Coulomb are unitless quantities related to proton and neutron numbers. The Optimal Constant Baseline uses the mode for categorical and mean for numerical properties.

keV to several MeV using hand-engineered inputs and considerable domain knowledge (Gao et al., 2021; Wang et al., 2014; Zeng et al., 2022; Wang et al.; Wu et al., 2022). Additionally, we provide a Tabular Denoising Diffusion Model (TDDPM) baseline from Kotelnikov et al. (2023). TDDPM is specifically designed to work on tabular data and only handles categorical and numerical features (omitting text, for instance). We evaluate models using 5 initialization seeds. See Appendix E.1 for more details. KBFormer has favorable performance on most properties (Table 2), but because TDDPM does not handle missing data naturally, its performance on the Stability property is not much better than the constant baseline. Finally, the probabilistic predictions enable reporting modeling uncertainties, which is critical for physics. We can use the denoising model to estimate various joint and conditional probabilities. Figure 11 in the appendix shows example binding energy uncertainty estimates. We scan intervals around the maximum likelihood obtained from the model on the evaluation set of binding energies. Future work will explore prediction uncertainties and implications.

**Generative Modeling for Tabular Data** In this section, we evaluate the quality of KBFormer generated samples via the performance of a downstream model trained on the synthetic data. Following Kotelnikov et al. (2023), we trained and tuned a GBDT to perform the downstream task, which can be either regression (evaluated with $R^2$) or classification (evaluated with $F_1$ score). Across 15 datasets, detailed in Appendix D, we generate 5 synthetic samples and train 10 GBDTs with different seeds. Then we evaluate the performance on a held-out test set from the original data. We compare performance against various generative models specializing in structured tabular data such as TabDDPM (Kotelnikov et al., 2023), CTAB-GAN (Zhao et al., 2021), TVAE (Xu et al., 2019) as well as an interpolation technique, SMOTE (Chawla et al., 2002), as a sanity check. Evaluation metrics as well as pre-processing are taken from Kotelnikov et al. (2023). In a majority of cases, KBFormer offers favorable performance, as shown in Table 3, but falls somewhat short on, notably, the largest dataset here: FB-Comments.

## 6 DISCUSSION AND CONCLUSION

KBFormer is a generative model of structured entities with potential in scientific modeling and KB completion, yet faces limitations that warrant further research (see Appendix G). Scaling the model to larger, more varied datasets poses significant challenges, particularly in large-scale pre-training. Integrating KBFormer with language models promises improvements in knowledge-intensive tasks, but requires exploration, especially in leveraging relationships within knowledge graphs for bet-

| | Method | ABAL $(R^2)$ | ADUL $(F_1)$ | BUDD $(F_1)$ | CALI $(R^2)$ | CARD $(F_1)$ | CHUR $(F_1)$ | DIAB $(F_1)$ |
|---|---|---|---|---|---|---|---|---|
| 0 | Real | $0.556_{\pm0.004}$ | $0.815_{\pm0.002}$ | $\mathbf{0.906_{\pm0.002}}$ | $0.857_{\pm0.001}$ | $0.738_{\pm0.001}$ | $0.740_{\pm0.009}$ | $0.785_{\pm0.013}$ |
| 1 | KBFormer | $\mathbf{0.550_{\pm0.009}}$ | $\mathbf{0.800_{\pm0.002}}$ | $\mathbf{0.907_{\pm0.003}}$ | $\mathbf{0.842_{\pm0.002}}$ | $\mathbf{0.736_{\pm0.001}}$ | $\mathbf{0.742_{\pm0.006}}$ | $\mathbf{0.763_{\pm0.016}}$ |
| 2 | TDDPM | $\mathbf{0.550_{\pm0.010}}$ | $0.795_{\pm0.001}$ | $\mathbf{0.906_{\pm0.003}}$ | $0.836_{\pm0.002}$ | $\mathbf{0.737_{\pm0.001}}$ | $\mathbf{0.755_{\pm0.006}}$ | $0.740_{\pm0.020}$ |
| 3 | SMOTE | $\mathbf{0.549_{\pm0.005}}$ | $0.791_{\pm0.002}$ | $0.891_{\pm0.003}$ | $\mathbf{0.840_{\pm0.001}}$ | $0.732_{\pm0.001}$ | $\mathbf{0.743_{\pm0.005}}$ | $0.683_{\pm0.037}$ |
| 4 | CTAB-GAN+ | $0.467_{\pm0.004}$ | $0.772_{\pm0.003}$ | $0.884_{\pm0.005}$ | $0.525_{\pm0.004}$ | $0.733_{\pm0.001}$ | $0.702_{\pm0.012}$ | $0.734_{\pm0.020}$ |
| 5 | CTAB-GAN | – | $0.783_{\pm0.002}$ | $0.855_{\pm0.005}$ | – | $0.717_{\pm0.001}$ | $0.688_{\pm0.006}$ | $0.731_{\pm0.022}$ |
| 6 | TVAE | $0.433_{\pm0.008}$ | $0.781_{\pm0.002}$ | $0.864_{\pm0.005}$ | $0.752_{\pm0.001}$ | $0.717_{\pm0.001}$ | $0.732_{\pm0.006}$ | $0.714_{\pm0.039}$ |

| | FB-C $(R^2)$ | GEST $(F_1)$ | HIGG $(F_1)$ | HOUS $(R^2)$ | INSU $(R^2)$ | KING $(R^2)$ | MINI $(F_1)$ | WILT $(F_1)$ |
|---|---|---|---|---|---|---|---|---|
| 0 | $0.837_{\pm0.001}$ | $0.636_{\pm0.007}$ | $0.724_{\pm0.001}$ | $0.662_{\pm0.003}$ | $0.814_{\pm0.001}$ | $0.907_{\pm0.002}$ | $0.934_{\pm0.000}$ | $0.898_{\pm0.006}$ |
| 1 | $0.687_{\pm0.004}$ | $0.605_{\pm0.008}$ | $\mathbf{0.721_{\pm0.001}}$ | $0.624_{\pm0.005}$ | $\mathbf{0.820_{\pm0.003}}$ | $\mathbf{0.876_{\pm0.006}}$ | $0.926_{\pm0.001}$ | $\mathbf{0.892_{\pm0.007}}$ |
| 2 | $0.713_{\pm0.002}$ | $0.597_{\pm0.006}$ | $\mathbf{0.722_{\pm0.001}}$ | $\mathbf{0.677_{\pm0.010}}$ | $0.809_{\pm0.002}$ | $0.833_{\pm0.014}$ | $\mathbf{0.936_{\pm0.001}}$ | $\mathbf{0.904_{\pm0.009}}$ |
| 3 | $\mathbf{0.803_{\pm0.002}}$ | $\mathbf{0.658_{\pm0.007}}$ | $\mathbf{0.722_{\pm0.001}}$ | $0.662_{\pm0.004}$ | $\mathbf{0.812_{\pm0.002}}$ | $0.842_{\pm0.004}$ | $0.932_{\pm0.001}$ | $\mathbf{0.913_{\pm0.007}}$ |
| 4 | $0.509_{\pm0.011}$ | $0.406_{\pm0.009}$ | $0.664_{\pm0.002}$ | $0.504_{\pm0.005}$ | $0.797_{\pm0.005}$ | $0.444_{\pm0.014}$ | $0.892_{\pm0.002}$ | $0.798_{\pm0.021}$ |
| 5 | – | $0.392_{\pm0.006}$ | $0.575_{\pm0.004}$ | – | – | – | $0.889_{\pm0.002}$ | $0.906_{\pm0.019}$ |
| 6 | $0.685_{\pm0.003}$ | $0.434_{\pm0.006}$ | $0.638_{\pm0.003}$ | $0.493_{\pm0.006}$ | $0.784_{\pm0.010}$ | $0.824_{\pm0.003}$ | $0.912_{\pm0.001}$ | $0.501_{\pm0.012}$ |

Table 3: Performance of a GBDT model trained on a downstream task on data generated by different tabular generative models (and on real data for comparison). Runs are averaged across 5 synthetic datasets and 10 GBDT training runs. Dashes denote results worse than the optimal constant solution.

ter generalization. In this work, we show applications of our approach to scientific modeling and KB completion for heterogeneous data types. The probabilistic nature of the model and its high-precision predictions for all numerical types make it suitable for a range of tasks. Another strong appeal of this approach is that it can learn latent representations of entities and their properties that other foundation models, such as LLMs, can attend over. This creates the opportunity to incorporate KBFormer in multimodal settings where structured data is one of the modalities. For example, future work may explore jointly training a combination of a KBFormer and an LLM for tasks like structured entity extraction from text and KB-augmented text generation. Unlike KB completion, for structured entity extraction the KBFormer would need to predict the entity properties based on LLM's latent representation of text, rather than unmasked properties. For KB-augmented text generation tasks, such as question-answering, it is the LLM that may attend over the latent representations of entities and properties from KBFormer. The ability to employ the same KBFormer model to these varied tasks opens up the opportunity to explore large-scale multitask pre-training of KBFormer, a potential stepping stone towards foundation models of structured entities and KBs. While large-scale KBs with structured entities are already available for pre-training, the ability to extract more structured information from text (and other modalities) creates a virtuous cycle by producing more data that may be employed for the training of KBFormer.

Existing foundation models, such as LLMs, store knowledge extracted from training data in their latent weights. This is undesirable for many reasons, including the fact that the stored knowledge is neither human-interpretable nor editable. To address this, Dai et al. (2022a) propose to isolate the neurons of the LLM to which specific facts may be approximately attributable and may even be editable. A more principled approach may involve a stricter separation-of-responsibility between parts of the model that is responsible for modeling language and that stores knowledge (Dai et al., 2022b). Alternatively, the knowledge store may simply be a search system that can retrieve relevant information from a corpus (Guu et al., 2020). A longer-term motivation for our current work is to develop models of structured knowledge that can augment LLMs and other similar models during pre-training. In such a design, a combination of KBFormer and and an exisitng LLM can be first used to extract structured data from a text corpus and then the combination of the extracted structured data and KBFromer can serve as an external knowledge store that a fresh LLM may read from during its pre-training. The fact that the KBFormer both produces and operates on explicitly human-interpretable structured data means that the learnt knowledge in this setting is amenable to both human inspection and curation. Our current work is a stepping stone towards that research vision.

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

## A  DIFFUSION LOSS

### A.1  A SIMPLE SIMULATION OF THE REVERSE PROCESS

The general form of the reverse process is as follows

$$\hat{R}_t(\mathbf{x}, \tilde{\mathbf{x}}) = \sum_{d=1}^{D} \delta(\mathbf{x}^{\neg d}, \tilde{\mathbf{x}}^{\neg d}) \hat{R}_t^d(\mathbf{x}, \tilde{x}^d), \tag{6}$$

where

$$\hat{R}_t^d(\mathbf{x}, \tilde{x}^d) = R_t^d(\tilde{x}^d, x^d) \sum_{x_0^d} p_{0|t}^\theta(x_0^d | \mathbf{x}^{\neg d}) \frac{q_{t|0}(\tilde{x}^d | x_0^d)}{q_{t|0}(x^d | x_0^d)} \tag{7}$$

The term $\hat{R}_t^d(\mathbf{x}, \tilde{x}^d)$ denotes the rate of events in the $d$-th dimension given the current state. Note that if the forward rate from $\tilde{x}^d$ to $x^d$ is zero, then the reverse rate from $x^d$ to $\tilde{x}^d$ will also be zero. For our setup, events can only occur out of the masking state in the reverse process, and all of the other states are now absorbing. Substituting the above equations for the rate and marginals of the absorbing process, we have

$$\hat{R}_t^d\left(\mathbf{x}, \tilde{x}^d\right) = \begin{cases} 0, & x^d \neq 0 \\ \beta(t) \dfrac{e^{-\gamma(t)}}{1 - e^{-\gamma(t)}} p_{0|t}^\theta\left(\tilde{x}^d \mid \mathbf{x}\right), & x^d = 0, \tilde{x}^d \neq 0 \\ -\beta(t) \dfrac{e^{-\gamma(t)}}{1 - e^{-\gamma(t)}}, & x^d = \tilde{x}^d = 0. \end{cases} \tag{8}$$

### A.2 LIKELIHOOD BOUND

The choice of an absorbing state kernel enables a simplified expression for the loss function with which the network can be trained. The general form of the ELBO (up to a constant) given by Campbell et al. (2022) is

$$\mathcal{L}_{CT} = \mathbb{E}_{t \sim \mathcal{U}(0,1) q_t(\mathbf{x}) r_t(\tilde{\mathbf{x}}|\mathbf{x})} \left[ \left\{ \sum_{\mathbf{x}' \neq \mathbf{x}} \hat{R}_t\left(\mathbf{x}, \mathbf{x}'\right) \right\} - Z_t(\mathbf{x}) \log \hat{R}\left(\tilde{\mathbf{x}}, \mathbf{x}\right) \right]. \tag{9}$$

The absorbing state setup enables two simplifications to this bound. First, we substitute in the form of $Z_t$ and $\hat{R}_t$ to obtain

$$\mathcal{L}_{CT} = \mathbb{E}_{t \sim \mathcal{U}(0,1) q_t(\mathbf{x}) r_t(\tilde{\mathbf{x}}|\mathbf{x})} \left[ \left\{ N_t \beta(t) \frac{e^{-\gamma(t)}}{1 - e^{-\gamma(t)}} \right\} - (\beta(t)(N_t - D)) \log \hat{R}\left(\tilde{\mathbf{x}}, \mathbf{x}\right) \right]. \tag{10}$$

This substitution has made the first term inside the expectation independent of the state and so omits the need for an additional pass of the neural network. Although Campbell et al. (2022) proposed to use a single pass of the neural net to give a good approximation of the bound, this formulation alleviates the need for that approximation.

Consider now the simulation of $q_t(\mathbf{x}) r_t(\tilde{\mathbf{x}}|\mathbf{x})$. Like Campbell et al. (2022) we simulate from the marginal of $\tilde{\mathbf{x}}$ and analytically marginalize the state $\mathbf{x}$. Since we know that in the forward process, all events occur at different times and that each event consists of flipping one property into the mask state (at the same rate across properties), simulating from the marginal $\psi(\tilde{\mathbf{x}}) = \sum_{\mathbf{x}} q_t(\mathbf{x}) r_t(\tilde{\mathbf{x}}|\mathbf{x})$ can be done by first masking out each property independently with probability $1 - \exp(-\gamma(t))$, and then masking out one additional property at random. In the case where all properties become masked by chance, we ignore this sample because $Z_t = 0$.

Having sampled from this marginal, consider the conditional state distribution $q_t(\mathbf{x}|\tilde{\mathbf{x}})$ the state $\mathbf{x}$ must have exactly one less mask than the state $\tilde{\mathbf{x}}$, uniformly at random. So analytically marginalizing this state leads to:

$$\mathcal{L}_{CT} = \mathbb{E}_{t \sim \mathcal{U}(0,1) \psi(\tilde{\mathbf{x}}) q_t(\mathbf{x}|\tilde{\mathbf{x}})} \left[ \left\{ N_t \beta(t) \frac{e^{-\gamma(t)}}{1 - e^{-\gamma(t)}} \right\} - (\beta(t)(N_t - D)) \log \hat{R}\left(\tilde{\mathbf{x}}, \mathbf{x}\right) \right] \tag{11}$$

$$= \mathbb{E}_{t \sim \mathcal{U}(0,1) \psi(\tilde{\mathbf{x}})} \left[ \left\{ N_t \beta(t) \frac{e^{-\gamma(t)}}{1 - e^{-\gamma(t)}} \right\} - \frac{\beta(t)(N_t - D)}{N_t + 1} \sum_{d | \tilde{x}^d = 0} \log \hat{R}\left(\tilde{\mathbf{x}}, x^d\right) \right]. \tag{12}$$

Note that sample $\tilde{\mathbf{x}}$ has $N_t + 1$ masked dimensions. We can now make our final substitution using equation 8 to get

$$\mathcal{L}_{CT} = \mathbb{E}_{t \sim \mathcal{U}(0,1)\psi(\tilde{\mathbf{x}})} \left[ \left\{ N_t \beta(t) \frac{e^{-\gamma(t)}}{1 - e^{-\gamma(t)}} \right\} - \frac{\beta(t)(N_t - D)}{N_t + 1} \sum_{d | \tilde{x}^d = 0} \log \beta(t) \frac{e^{-\gamma(t)}}{1 - e^{-\gamma(t)}} p_{0|t}^\theta \left( \tilde{x}^d \mid \mathbf{x} \right) \right] . \tag{13}$$

Dropping terms that do not depend on neural network parameters we obtain

$$\mathcal{L}_{CT} = \mathbb{E}_{t \sim \mathcal{U}(0,1)\psi(\tilde{\mathbf{x}})} \left[ -\frac{\beta(t)(N_t - D)}{N_t + 1} \sum_{d | \tilde{x}^d = 0} \log p_{0|t}^\theta \left( \tilde{x}^d \mid \mathbf{x} \right) + \text{const} \right] . \tag{14}$$

Finally, we can change the variable of integration from $t$ to the probability of flipping a property in to the mask state. Writing $\pi(t) = 1 - \exp(-\gamma(t))$, we have

$$\frac{d\pi}{dt} = \frac{d\gamma}{dt} e^{-\gamma(t)} = \beta(t)(1 - \pi(t)), \tag{15}$$

and so the objective becomes

$$\mathcal{L}_{CT} = \mathbb{E}_{\pi \sim \mathcal{U}(0,1)\psi(\tilde{\mathbf{x}})} \left[ \frac{1 - \hat{\pi}}{1 - \pi} \frac{D}{N_t + 1} \sum_{d | \tilde{x}^d = 0} \log p_{0|t}^\theta \left( \tilde{x}^d \mid \mathbf{x} \right) + \text{const} \right] , \tag{16}$$

where we use $\hat{\pi} = N_t / D$ as the empirical masking rate.

This final simplification of the objective reveals a close connection to self-supervised learning: we have the standard reconstruction loss for randomly masked elements in $x_0$, but with a random amount of masking. The factor $(1 - \hat{\pi})/(1 - \pi)$ is the ratio of non-masked elements to the expected non-masked elements, so will downweight gradients where the amount of information is less than expected *i.e.*, if by chance, more masked are flipped than $\pi$ would imply, then the sample is down-weighted.

# B ABLATIONS

## B.1 AUTOREGRESSIVE DIFFUSION VS. MASKED MODELING

### B.1.1 QUANTITITAVE EXAMPLES

Here, we show how our prescription for generating samples from our autoregressive diffusion process compares against simple masked modeling where all masked properties are predicted simultaneously. We show a few qualitative examples, including visually inspecting generated MNIST samples (Appendix B.1.2). In this section, we will make this intuition more quantitative by using our tabular datasets testbed.

We use the same model trained with a random masking rate to generate samples using each approach on each dataset. Then we compare the performance of a downstream model trained on these synthetic samples. The results are shown in Table 4. Unsurprisingly, diffusion-based sampling outperforms masked modeling in all cases. We conjecture the wide variance across tasks in the non-diffusion case to the importance of correlations across features in each dataset. Indeed, if the properties are completely independent, it suffices to sample from the marginals, of which the non-autoregressive model is perfectly capable. However, if there are strong correlations, sampling from the marginals can lead to completely smoothed-out samples, which results in performance no better than a constant baseline.

### B.1.2 QUALITATIVE EXAMPLES

For a more visual representation of our generative model we train a simple U-Net to generate MNIST images starting from a blank image (fully masked) using an implementation of the reverse process from Appendix A.1. Here we show a few examples conditioned on the digit label.

| | Method | ABAL $(R^2)$ | ADUL $(F_1)$ | BUDD $(F_1)$ | CALI $(R^2)$ | CARD $(F_1)$ | CHUR $(F_1)$ | DIAB $(F_1)$ |
|---|---|---|---|---|---|---|---|---|
| 0 | Real | $0.556_{\pm0.004}$ | $0.815_{\pm0.002}$ | $0.906_{\pm0.002}$ | $0.857_{\pm0.001}$ | $0.738_{\pm0.001}$ | $0.740_{\pm0.009}$ | $0.785_{\pm0.013}$ |
| 1 | KBFormer | $\mathbf{0.550}_{\pm\mathbf{0.009}}$ | $\mathbf{0.800}_{\pm\mathbf{0.002}}$ | $\mathbf{0.907}_{\pm\mathbf{0.003}}$ | $\mathbf{0.842}_{\pm\mathbf{0.002}}$ | $\mathbf{0.736}_{\pm\mathbf{0.001}}$ | $\mathbf{0.742}_{\pm\mathbf{0.006}}$ | $\mathbf{0.763}_{\pm\mathbf{0.016}}$ |
| 2 | Single-step KBFormer | $0.027_{\pm0.041}$ | $0.776_{\pm0.006}$ | – | $0.001_{\pm0.003}$ | $0.730_{\pm0.001}$ | $0.711_{\pm0.005}$ | $0.730_{\pm0.021}$ |

| | FB-C $(R^2)$ | GEST $(F_1)$ | HIGG $(F_1)$ | HOUS $(R^2)$ | INSU $(R^2)$ | KING $(R^2)$ | MINI $(F_1)$ | WILT $(F_1)$ |
|---|---|---|---|---|---|---|---|---|
| 0 | $0.837_{\pm0.001}$ | $0.636_{\pm0.007}$ | $0.724_{\pm0.001}$ | $0.662_{\pm0.003}$ | $0.814_{\pm0.001}$ | $0.907_{\pm0.002}$ | $0.934_{\pm0.000}$ | $0.898_{\pm0.006}$ |
| 1 | $\mathbf{0.687}_{\pm\mathbf{0.004}}$ | $\mathbf{0.605}_{\pm\mathbf{0.008}}$ | $\mathbf{0.721}_{\pm\mathbf{0.001}}$ | $\mathbf{0.624}_{\pm\mathbf{0.005}}$ | $\mathbf{0.820}_{\pm\mathbf{0.003}}$ | $\mathbf{0.876}_{\pm\mathbf{0.006}}$ | $\mathbf{0.926}_{\pm\mathbf{0.001}}$ | $\mathbf{0.892}_{\pm\mathbf{0.007}}$ |
| 2 | $0.095_{\pm0.025}$ | $0.434_{\pm0.009}$ | – | $0.002_{\pm0.006}$ | $-0.002_{\pm0.013}$ | $0.071_{\pm0.050}$ | $0.834_{\pm0.002}$ | $0.562_{\pm0.031}$ |

Table 4: Ablating our autoregressive diffusion (Row 1) against simple masked modeling (Row 2) where all properties are predicted simultaneously.

Binary MNIST images are generated by treating pixels as binary categorical variables and diffusing through pixel space one at a time. As Figure 6 illustrates, diffusion generates coherent sample digits emerging through gradual reveals. In contrast, (non-autoregressive) masked modeling exposes all pixels at once, lacking the proper correlations, evident by the noisy samples. While autoregressive benefits are well-established, this visualization demonstrates that diffusion more accurately captures relationships during entity generation than simple masked modeling.

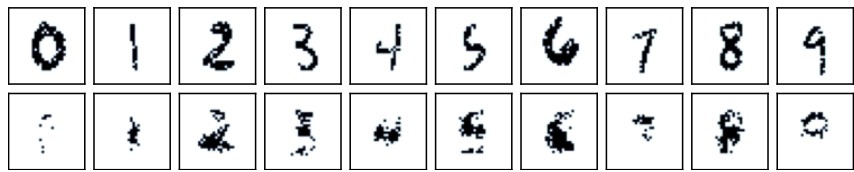

Figure 6: Class-conditioned MNIST samples utilizing (top) a pixel-by-pixel discrete diffusion, or (bottom) unveiling the entire image simultaneously through masked modeling.

Another qualitative example of samples from a KBFormer model trained on the GSMArena dataset (see Section 4) highlighting the benefits of this approach in capturing multimodality is shown in Figure 7.

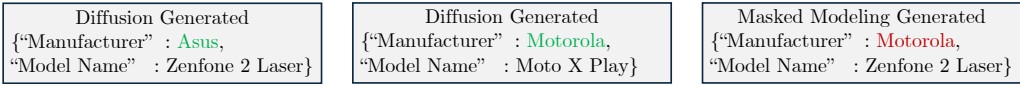

Figure 7: Diffusion samples from KBFormer yield consistent model names and manufacturers, while masked modeling mismatches manufacturers and names by only capturing marginals.

## B.2 ABLATIONS FOR PREDICTION QUALITY ON GSM

In this section we run several ablations on different model choices, training on the GSMArena dataset and evaluating with respect to RMS, accuracy and word IOU (compare with Table 1). Overall we find that the models performance is quite stable with respect to the changes we introduce. We include experiments for two different hidden dimensions, two different learning rates, periodic and DICE embeddings, number of GMM mixtures and whether or not numerical embeddings are shared across properties. Table 5 provides the data on all the experiments we ran for this section. We break up interesting aspects into smaller tables for better overview. The performance for each setting is averaged over 5 model initialization seeds.

**Numerical Embeddings** The first ablation pertains to the treatment of numerical values on the encoder side. Two options are provided for the way in which we embed numerical values, via DICE (Sundararaman et al., 2020) and via periodic embeddings (Vaswani et al., 2017). Additionally, we look at whether tying the embeddings or all nuclear properties has an effect on performance. We

keep other hyperparameters fixed: Notably we run with a model dimension of 512, 50 GMM mixtures for each numerical property, a learning rate of 0.001 and a random mask rate during training. The results are shown in Table 6. They are overall comparable, differences in performance for any field are within one standard deviation. Interestingly, the uncertainty over different initializations seems generally smaller when numerical embeddings are tied.

**GMM vs MSE**   In this ablation we vary treatment of numerical properties on the output side. In Figure 5 we illustrate the benefit of using GMMs as opposed to a simple MSE regression, specifically for generation quality. Here, we investigate this choice with respect to prediction quality Experiments are shown in Table 7. The results align with our expectation of similar performance. The reason for this is that in a regression task, we predict the value value as the weighted sum of the mean values of all mixtures, which should attain similar performance as fitting only one Gaussian and predicting its mean.

**Masking rate**   During training of a KBFormer model, properties are masked out at random. In the scheme derived in Section 3.2 and Appendix A, the rate at which properties are masked is also chosen uniformly at random from 0 to 1. Here, we explore how prediction quality changes when training with a fixed masking rate of 0.5 instead. The results can be seen in Table 8. Interestingly, we seem to find a small but fairly consistent difference in performance. Except for the text field "model", in which the performance is significantly better, the other predictions are slightly worse when training with the constant masking rate. In future work, we will explore this apparent trade-off further and hope to find how and where the model treats text fields differently than categorical and numerical fields.

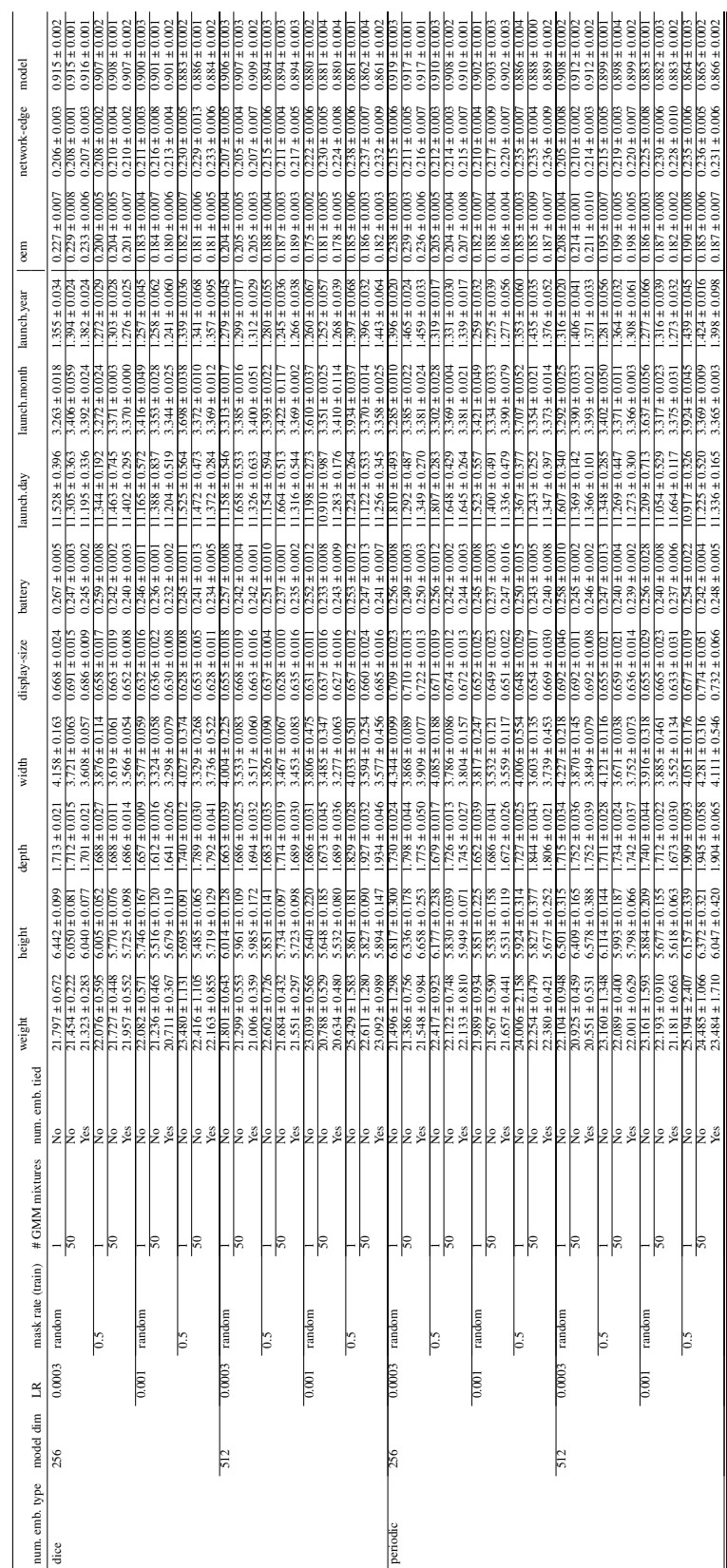

| num. emb. type | model dim | LR | mask rate (train) | # GMM mixtures | num. emb. tied | weight | height | depth | width | display-size | battery | launch.day | launch.month | launch.year | oem | network-edge | model |
|---|---|---|---|---|---|---|---|---|---|---|---|---|---|---|---|---|---|
| dice | 256 | 0.0003 | random | 1 | No | 21.797 ± 0.672 | 6.442 ± 0.099 | 1.713 ± 0.021 | 4.158 ± 0.163 | 0.668 ± 0.024 | 0.267 ± 0.005 | 11.528 ± 0.396 | 3.263 ± 0.018 | 1.355 ± 0.034 | 0.227 ± 0.007 | 0.206 ± 0.003 | 0.915 ± 0.002 |
| | | | | 50 | No | 21.454 ± 0.222 | 6.050 ± 0.081 | 1.712 ± 0.015 | 3.721 ± 0.063 | 0.691 ± 0.015 | 0.247 ± 0.003 | 11.305 ± 0.363 | 3.406 ± 0.059 | 1.394 ± 0.024 | 0.229 ± 0.008 | 0.208 ± 0.001 | 0.915 ± 0.001 |
| | | | | 50 | Yes | 21.323 ± 0.283 | 6.040 ± 0.077 | 1.701 ± 0.021 | 3.608 ± 0.057 | 0.686 ± 0.009 | 0.245 ± 0.002 | 11.195 ± 0.336 | 3.392 ± 0.024 | 1.382 ± 0.024 | 0.233 ± 0.006 | 0.207 ± 0.003 | 0.916 ± 0.001 |
| | | | 0.5 | 1 | No | 22.076 ± 0.595 | 6.005 ± 0.052 | 1.688 ± 0.027 | 3.876 ± 0.114 | 0.658 ± 0.017 | 0.259 ± 0.008 | 11.344 ± 0.192 | 3.272 ± 0.024 | 1.272 ± 0.029 | 0.200 ± 0.005 | 0.208 ± 0.002 | 0.907 ± 0.002 |
| | | | | 50 | No | 21.727 ± 0.448 | 5.770 ± 0.076 | 1.688 ± 0.011 | 3.619 ± 0.061 | 0.663 ± 0.019 | 0.242 ± 0.002 | 11.463 ± 0.745 | 3.371 ± 0.003 | 1.303 ± 0.028 | 0.204 ± 0.005 | 0.210 ± 0.004 | 0.908 ± 0.001 |
| | | | | 50 | Yes | 21.957 ± 0.552 | 5.723 ± 0.098 | 1.686 ± 0.014 | 3.566 ± 0.054 | 0.652 ± 0.008 | 0.240 ± 0.003 | 11.402 ± 0.295 | 3.370 ± 0.000 | 1.276 ± 0.025 | 0.201 ± 0.007 | 0.210 ± 0.002 | 0.907 ± 0.002 |
| | | 0.001 | random | 1 | No | 22.082 ± 0.571 | 5.746 ± 0.167 | 1.657 ± 0.009 | 3.577 ± 0.059 | 0.632 ± 0.016 | 0.246 ± 0.011 | 11.165 ± 0.572 | 3.416 ± 0.049 | 1.257 ± 0.045 | 0.183 ± 0.004 | 0.211 ± 0.003 | 0.900 ± 0.003 |
| | | | | 50 | No | 21.236 ± 0.465 | 5.516 ± 0.120 | 1.612 ± 0.016 | 3.324 ± 0.058 | 0.636 ± 0.022 | 0.236 ± 0.001 | 11.388 ± 0.837 | 3.353 ± 0.028 | 1.258 ± 0.062 | 0.184 ± 0.007 | 0.216 ± 0.008 | 0.901 ± 0.001 |
| | | | | 50 | Yes | 20.711 ± 0.367 | 5.679 ± 0.119 | 1.641 ± 0.026 | 3.298 ± 0.079 | 0.630 ± 0.008 | 0.232 ± 0.002 | 11.204 ± 0.519 | 3.344 ± 0.025 | 1.241 ± 0.060 | 0.180 ± 0.006 | 0.213 ± 0.004 | 0.901 ± 0.002 |
| | | | 0.5 | 1 | No | 23.480 ± 1.731 | 5.695 ± 0.091 | 1.740 ± 0.012 | 4.027 ± 0.374 | 0.628 ± 0.008 | 0.245 ± 0.011 | 11.525 ± 0.564 | 3.698 ± 0.038 | 1.339 ± 0.036 | 0.182 ± 0.007 | 0.230 ± 0.005 | 0.883 ± 0.002 |
| | | | | 50 | No | 22.416 ± 1.105 | 5.485 ± 0.065 | 1.789 ± 0.030 | 3.329 ± 0.268 | 0.653 ± 0.005 | 0.241 ± 0.013 | 11.472 ± 0.473 | 3.372 ± 0.010 | 1.341 ± 0.068 | 0.181 ± 0.006 | 0.229 ± 0.013 | 0.886 ± 0.001 |
| | | | | 50 | Yes | 22.163 ± 0.855 | 5.719 ± 0.129 | 1.792 ± 0.041 | 3.736 ± 0.522 | 0.628 ± 0.011 | 0.234 ± 0.005 | 11.372 ± 0.284 | 3.369 ± 0.012 | 1.357 ± 0.095 | 0.181 ± 0.005 | 0.233 ± 0.006 | 0.884 ± 0.002 |
| | 512 | 0.0003 | random | 1 | No | 21.801 ± 0.643 | 6.014 ± 0.128 | 1.663 ± 0.039 | 4.004 ± 0.225 | 0.655 ± 0.018 | 0.257 ± 0.008 | 11.158 ± 0.546 | 3.313 ± 0.017 | 1.279 ± 0.045 | 0.204 ± 0.004 | 0.207 ± 0.005 | 0.906 ± 0.003 |
| | | | | 50 | No | 21.299 ± 0.553 | 5.961 ± 0.109 | 1.686 ± 0.025 | 3.533 ± 0.083 | 0.668 ± 0.019 | 0.242 ± 0.004 | 11.658 ± 0.533 | 3.385 ± 0.016 | 1.299 ± 0.017 | 0.205 ± 0.005 | 0.205 ± 0.004 | 0.907 ± 0.003 |
| | | | | 50 | Yes | 21.006 ± 0.359 | 5.985 ± 0.172 | 1.694 ± 0.032 | 3.517 ± 0.060 | 0.663 ± 0.016 | 0.242 ± 0.001 | 11.326 ± 0.633 | 3.400 ± 0.051 | 1.312 ± 0.029 | 0.205 ± 0.003 | 0.207 ± 0.007 | 0.909 ± 0.002 |
| | | | 0.5 | 1 | No | 22.602 ± 0.726 | 5.851 ± 0.141 | 1.683 ± 0.035 | 3.826 ± 0.090 | 0.637 ± 0.004 | 0.251 ± 0.010 | 11.154 ± 0.594 | 3.393 ± 0.022 | 1.280 ± 0.055 | 0.188 ± 0.004 | 0.215 ± 0.006 | 0.894 ± 0.003 |
| | | | | 50 | No | 21.684 ± 0.432 | 5.734 ± 0.097 | 1.714 ± 0.019 | 3.467 ± 0.067 | 0.628 ± 0.010 | 0.237 ± 0.001 | 11.664 ± 0.513 | 3.422 ± 0.117 | 1.245 ± 0.036 | 0.187 ± 0.003 | 0.211 ± 0.004 | 0.894 ± 0.003 |
| | | | | 50 | Yes | 21.551 ± 0.297 | 5.723 ± 0.098 | 1.689 ± 0.030 | 3.453 ± 0.083 | 0.635 ± 0.016 | 0.235 ± 0.002 | 11.316 ± 0.544 | 3.369 ± 0.002 | 1.266 ± 0.038 | 0.189 ± 0.003 | 0.217 ± 0.005 | 0.894 ± 0.003 |
| | | 0.001 | random | 1 | No | 23.039 ± 0.565 | 5.640 ± 0.220 | 1.686 ± 0.031 | 3.806 ± 0.475 | 0.631 ± 0.011 | 0.252 ± 0.012 | 11.198 ± 0.273 | 3.610 ± 0.037 | 1.260 ± 0.067 | 0.175 ± 0.002 | 0.222 ± 0.006 | 0.880 ± 0.002 |
| | | | | 50 | No | 20.788 ± 0.529 | 5.648 ± 0.185 | 1.673 ± 0.045 | 3.485 ± 0.347 | 0.637 ± 0.016 | 0.233 ± 0.008 | 10.910 ± 0.987 | 3.351 ± 0.025 | 1.252 ± 0.057 | 0.181 ± 0.005 | 0.230 ± 0.005 | 0.881 ± 0.004 |
| | | | | 50 | Yes | 20.634 ± 0.480 | 5.532 ± 0.080 | 1.689 ± 0.036 | 3.277 ± 0.063 | 0.627 ± 0.016 | 0.243 ± 0.009 | 11.283 ± 0.176 | 3.410 ± 0.114 | 1.268 ± 0.039 | 0.178 ± 0.005 | 0.224 ± 0.008 | 0.880 ± 0.004 |
| | | | 0.5 | 1 | No | 25.429 ± 1.583 | 5.861 ± 0.181 | 1.829 ± 0.028 | 4.033 ± 0.501 | 0.657 ± 0.012 | 0.253 ± 0.012 | 11.224 ± 0.264 | 3.934 ± 0.037 | 1.397 ± 0.068 | 0.185 ± 0.006 | 0.238 ± 0.006 | 0.861 ± 0.001 |
| | | | | 50 | No | 22.611 ± 1.280 | 5.827 ± 0.090 | 1.927 ± 0.032 | 3.594 ± 0.254 | 0.660 ± 0.024 | 0.247 ± 0.013 | 11.122 ± 0.533 | 3.370 ± 0.014 | 1.396 ± 0.032 | 0.186 ± 0.003 | 0.237 ± 0.007 | 0.862 ± 0.004 |
| | | | | 50 | Yes | 23.092 ± 0.989 | 5.894 ± 0.147 | 1.934 ± 0.046 | 3.573 ± 0.456 | 0.685 ± 0.016 | 0.241 ± 0.007 | 11.256 ± 0.345 | 3.358 ± 0.025 | 1.443 ± 0.064 | 0.182 ± 0.003 | 0.232 ± 0.009 | 0.861 ± 0.002 |
| periodic | 256 | 0.0003 | random | 1 | No | 21.496 ± 1.298 | 6.817 ± 0.300 | 1.730 ± 0.024 | 4.344 ± 0.099 | 0.709 ± 0.023 | 0.256 ± 0.008 | 11.810 ± 0.493 | 3.285 ± 0.010 | 1.396 ± 0.020 | 0.238 ± 0.003 | 0.215 ± 0.006 | 0.919 ± 0.003 |
| | | | | 50 | No | 21.386 ± 0.756 | 6.336 ± 0.178 | 1.798 ± 0.044 | 3.868 ± 0.089 | 0.710 ± 0.013 | 0.249 ± 0.003 | 11.292 ± 0.487 | 3.385 ± 0.022 | 1.465 ± 0.024 | 0.239 ± 0.003 | 0.211 ± 0.005 | 0.917 ± 0.001 |
| | | | | 50 | Yes | 21.548 ± 0.984 | 6.658 ± 0.253 | 1.775 ± 0.050 | 3.909 ± 0.077 | 0.722 ± 0.013 | 0.250 ± 0.003 | 11.349 ± 0.270 | 3.381 ± 0.024 | 1.459 ± 0.033 | 0.236 ± 0.006 | 0.216 ± 0.007 | 0.917 ± 0.001 |
| | | | 0.5 | 1 | No | 22.417 ± 0.923 | 6.177 ± 0.238 | 1.679 ± 0.017 | 4.085 ± 0.188 | 0.671 ± 0.019 | 0.256 ± 0.012 | 11.807 ± 0.283 | 3.302 ± 0.028 | 1.319 ± 0.017 | 0.205 ± 0.005 | 0.212 ± 0.003 | 0.910 ± 0.003 |
| | | | | 50 | No | 22.122 ± 0.748 | 5.830 ± 0.039 | 1.726 ± 0.013 | 3.786 ± 0.086 | 0.674 ± 0.012 | 0.242 ± 0.002 | 11.648 ± 0.429 | 3.369 ± 0.004 | 1.331 ± 0.030 | 0.204 ± 0.004 | 0.214 ± 0.003 | 0.908 ± 0.002 |
| | | | | 50 | Yes | 22.133 ± 0.810 | 5.949 ± 0.071 | 1.745 ± 0.027 | 3.804 ± 0.157 | 0.672 ± 0.013 | 0.244 ± 0.003 | 11.645 ± 0.264 | 3.381 ± 0.021 | 1.339 ± 0.017 | 0.207 ± 0.008 | 0.215 ± 0.007 | 0.910 ± 0.001 |
| | | 0.001 | random | 1 | No | 21.989 ± 0.934 | 5.851 ± 0.225 | 1.652 ± 0.039 | 3.817 ± 0.247 | 0.652 ± 0.025 | 0.245 ± 0.008 | 11.523 ± 0.557 | 3.421 ± 0.049 | 1.259 ± 0.032 | 0.182 ± 0.007 | 0.210 ± 0.004 | 0.902 ± 0.001 |
| | | | | 50 | No | 21.567 ± 0.590 | 5.538 ± 0.158 | 1.686 ± 0.041 | 3.532 ± 0.121 | 0.649 ± 0.023 | 0.237 ± 0.003 | 11.400 ± 0.491 | 3.334 ± 0.033 | 1.275 ± 0.039 | 0.188 ± 0.004 | 0.217 ± 0.009 | 0.903 ± 0.003 |
| | | | | 50 | Yes | 21.657 ± 0.441 | 5.531 ± 0.119 | 1.672 ± 0.026 | 3.559 ± 0.117 | 0.651 ± 0.022 | 0.247 ± 0.016 | 11.336 ± 0.479 | 3.390 ± 0.079 | 1.277 ± 0.056 | 0.186 ± 0.004 | 0.220 ± 0.007 | 0.902 ± 0.003 |
| | | | 0.5 | 1 | No | 24.006 ± 2.158 | 5.924 ± 0.314 | 1.727 ± 0.025 | 4.006 ± 0.554 | 0.648 ± 0.029 | 0.250 ± 0.015 | 11.367 ± 0.377 | 3.707 ± 0.052 | 1.355 ± 0.060 | 0.183 ± 0.003 | 0.235 ± 0.007 | 0.886 ± 0.004 |
| | | | | 50 | No | 22.254 ± 0.479 | 5.827 ± 0.377 | 1.844 ± 0.043 | 3.603 ± 0.135 | 0.654 ± 0.017 | 0.243 ± 0.005 | 11.243 ± 0.252 | 3.354 ± 0.021 | 1.435 ± 0.035 | 0.185 ± 0.009 | 0.235 ± 0.004 | 0.888 ± 0.000 |
| | | | | 50 | Yes | 22.380 ± 0.421 | 5.677 ± 0.252 | 1.806 ± 0.021 | 3.739 ± 0.453 | 0.669 ± 0.030 | 0.240 ± 0.008 | 11.347 ± 0.397 | 3.373 ± 0.014 | 1.376 ± 0.052 | 0.187 ± 0.007 | 0.236 ± 0.009 | 0.889 ± 0.002 |
| | 512 | 0.0003 | random | 1 | No | 22.104 ± 0.948 | 6.501 ± 0.315 | 1.715 ± 0.034 | 4.227 ± 0.218 | 0.692 ± 0.046 | 0.258 ± 0.010 | 11.607 ± 0.340 | 3.292 ± 0.025 | 1.316 ± 0.020 | 0.208 ± 0.004 | 0.205 ± 0.008 | 0.908 ± 0.002 |
| | | | | 50 | No | 20.925 ± 0.459 | 6.409 ± 0.165 | 1.752 ± 0.036 | 3.870 ± 0.145 | 0.692 ± 0.011 | 0.245 ± 0.002 | 11.369 ± 0.142 | 3.390 ± 0.033 | 1.406 ± 0.041 | 0.214 ± 0.001 | 0.210 ± 0.002 | 0.912 ± 0.002 |
| | | | | 50 | Yes | 20.551 ± 0.531 | 6.578 ± 0.388 | 1.752 ± 0.039 | 3.849 ± 0.079 | 0.692 ± 0.008 | 0.246 ± 0.002 | 11.366 ± 0.101 | 3.393 ± 0.021 | 1.371 ± 0.033 | 0.211 ± 0.010 | 0.214 ± 0.003 | 0.912 ± 0.002 |
| | | | 0.5 | 1 | No | 23.160 ± 1.348 | 6.114 ± 0.144 | 1.711 ± 0.028 | 4.121 ± 0.116 | 0.655 ± 0.021 | 0.247 ± 0.013 | 11.348 ± 0.285 | 3.402 ± 0.050 | 1.281 ± 0.056 | 0.195 ± 0.007 | 0.215 ± 0.005 | 0.899 ± 0.001 |
| | | | | 50 | No | 22.089 ± 0.400 | 5.993 ± 0.187 | 1.734 ± 0.024 | 3.671 ± 0.038 | 0.659 ± 0.021 | 0.240 ± 0.004 | 11.269 ± 0.447 | 3.371 ± 0.011 | 1.364 ± 0.032 | 0.199 ± 0.005 | 0.219 ± 0.003 | 0.898 ± 0.004 |
| | | | | 50 | Yes | 22.001 ± 0.629 | 5.798 ± 0.066 | 1.742 ± 0.037 | 3.752 ± 0.073 | 0.636 ± 0.014 | 0.239 ± 0.002 | 11.273 ± 0.300 | 3.366 ± 0.003 | 1.308 ± 0.061 | 0.198 ± 0.005 | 0.220 ± 0.007 | 0.899 ± 0.002 |
| | | 0.001 | random | 1 | No | 23.161 ± 1.593 | 5.884 ± 0.209 | 1.740 ± 0.044 | 3.910 ± 0.318 | 0.655 ± 0.029 | 0.256 ± 0.028 | 11.209 ± 0.713 | 3.637 ± 0.056 | 1.277 ± 0.066 | 0.186 ± 0.003 | 0.225 ± 0.008 | 0.883 ± 0.001 |
| | | | | 50 | No | 22.193 ± 0.910 | 5.677 ± 0.155 | 1.712 ± 0.022 | 3.885 ± 0.461 | 0.665 ± 0.023 | 0.240 ± 0.008 | 11.054 ± 0.529 | 3.317 ± 0.023 | 1.316 ± 0.039 | 0.187 ± 0.008 | 0.230 ± 0.006 | 0.882 ± 0.003 |
| | | | | 50 | Yes | 21.181 ± 0.663 | 5.618 ± 0.063 | 1.673 ± 0.030 | 3.552 ± 0.134 | 0.633 ± 0.031 | 0.237 ± 0.006 | 11.664 ± 0.117 | 3.375 ± 0.031 | 1.273 ± 0.032 | 0.182 ± 0.002 | 0.228 ± 0.010 | 0.883 ± 0.002 |
| | | | 0.5 | 1 | No | 25.194 ± 2.407 | 6.157 ± 0.339 | 1.909 ± 0.093 | 4.051 ± 0.176 | 0.677 ± 0.019 | 0.254 ± 0.022 | 10.917 ± 0.326 | 3.924 ± 0.045 | 1.439 ± 0.045 | 0.190 ± 0.008 | 0.235 ± 0.006 | 0.864 ± 0.005 |
| | | | | 50 | No | 24.485 ± 1.066 | 6.372 ± 0.321 | 1.945 ± 0.058 | 4.281 ± 0.316 | 0.774 ± 0.051 | 0.242 ± 0.004 | 11.225 ± 0.520 | 3.369 ± 0.009 | 1.424 ± 0.016 | 0.185 ± 0.006 | 0.236 ± 0.005 | 0.865 ± 0.002 |
| | | | | 50 | Yes | 23.484 ± 1.710 | 6.047 ± 0.420 | 1.904 ± 0.065 | 4.111 ± 0.546 | 0.732 ± 0.066 | 0.248 ± 0.005 | 11.336 ± 0.165 | 3.365 ± 0.003 | 1.398 ± 0.098 | 0.187 ± 0.007 | 0.231 ± 0.006 | 0.866 ± 0.002 |

Table 5: Ablations over embedding types, mask rates number of mixtures in the GMMs and tied numerical embeddings.

| num. emb. type | DICE | | Periodic Embedding | |
|---|---|---|---|---|
| num. emb. tied | No | Yes | No | Yes |
| weight | 20.788 ± 0.529 | 20.634 ± 0.480 | 22.193 ± 0.910 | 21.181 ± 0.663 |
| height | 5.648 ± 0.185 | 5.532 ± 0.080 | 5.677 ± 0.155 | 5.618 ± 0.063 |
| depth | 1.673 ± 0.045 | 1.689 ± 0.036 | 1.712 ± 0.022 | 1.673 ± 0.030 |
| width | 3.485 ± 0.347 | 3.277 ± 0.063 | 3.885 ± 0.461 | 3.552 ± 0.134 |
| display-size | 0.637 ± 0.016 | 0.627 ± 0.016 | 0.665 ± 0.023 | 0.633 ± 0.031 |
| battery | 0.233 ± 0.008 | 0.243 ± 0.009 | 0.240 ± 0.008 | 0.237 ± 0.006 |
| launch.day | 10.910 ± 0.987 | 11.283 ± 0.176 | 11.054 ± 0.529 | 11.664 ± 0.117 |
| launch.month | 3.351 ± 0.025 | 3.410 ± 0.114 | 3.317 ± 0.023 | 3.375 ± 0.031 |
| launch.year | 1.252 ± 0.057 | 1.268 ± 0.039 | 1.316 ± 0.039 | 1.273 ± 0.032 |
| oem | 0.181 ± 0.005 | 0.178 ± 0.005 | 0.187 ± 0.008 | 0.182 ± 0.002 |
| network-edge | 0.230 ± 0.005 | 0.224 ± 0.008 | 0.230 ± 0.006 | 0.228 ± 0.010 |
| model | 0.881 ± 0.004 | 0.880 ± 0.004 | 0.882 ± 0.003 | 0.883 ± 0.002 |

Table 6: Property prediction performance of the KBFormer on the GSM dataset, with two different numerical embeddings, either tied or untied. In the tied case, the numerical embeddings are shared between all numerical properties. Runs are averaged over 5 model initialization seeds. Other hyperparameters are fixed.

| num. emb. type | DICE | | Periodic Embedding | |
|---|---|---|---|---|
| # GMM mixtures | 1 | 50 | 1 | 50 |
| weight | 23.039 ± 0.565 | 20.788 ± 0.529 | 23.161 ± 1.593 | 22.193 ± 0.910 |
| height | 5.640 ± 0.220 | 5.648 ± 0.185 | 5.884 ± 0.209 | 5.677 ± 0.155 |
| depth | 1.686 ± 0.031 | 1.673 ± 0.045 | 1.740 ± 0.044 | 1.712 ± 0.022 |
| width | 3.806 ± 0.475 | 3.485 ± 0.347 | 3.916 ± 0.318 | 3.885 ± 0.461 |
| display-size | 0.631 ± 0.011 | 0.637 ± 0.016 | 0.655 ± 0.029 | 0.665 ± 0.023 |
| battery | 0.252 ± 0.012 | 0.233 ± 0.008 | 0.256 ± 0.028 | 0.240 ± 0.008 |
| launch.day | 11.198 ± 0.273 | 10.910 ± 0.987 | 11.209 ± 0.713 | 11.054 ± 0.529 |
| launch.month | 3.610 ± 0.037 | 3.351 ± 0.025 | 3.637 ± 0.056 | 3.317 ± 0.023 |
| launch.year | 1.260 ± 0.067 | 1.252 ± 0.057 | 1.277 ± 0.066 | 1.316 ± 0.039 |
| oem | 0.175 ± 0.002 | 0.181 ± 0.005 | 0.186 ± 0.003 | 0.187 ± 0.008 |
| network-edge | 0.222 ± 0.006 | 0.230 ± 0.005 | 0.225 ± 0.008 | 0.230 ± 0.006 |
| model | 0.880 ± 0.002 | 0.881 ± 0.004 | 0.883 ± 0.001 | 0.882 ± 0.003 |

Table 7: Prediction performance on GSM with either 1 or 50 GMM mixtures per numerical property. When the number of GMM mixtures is 1, the task reduces to regression via MSE.

| num. emb. tied | No | | Yes | |
|---|---|---|---|---|
| mask rate (training) | Random | 0.5 | Random | 0.5 |
| weight | 20.788 ± 0.529 | 22.611 ± 1.280 | 20.634 ± 0.480 | 23.092 ± 0.989 |
| height | 5.648 ± 0.185 | 5.827 ± 0.090 | 5.532 ± 0.080 | 5.894 ± 0.147 |
| depth | 1.673 ± 0.045 | 1.927 ± 0.032 | 1.689 ± 0.036 | 1.934 ± 0.046 |
| width | 3.485 ± 0.347 | 3.594 ± 0.254 | 3.277 ± 0.063 | 3.577 ± 0.456 |
| display-size | 0.637 ± 0.016 | 0.660 ± 0.024 | 0.627 ± 0.016 | 0.685 ± 0.016 |
| battery | 0.233 ± 0.008 | 0.247 ± 0.013 | 0.243 ± 0.009 | 0.241 ± 0.007 |
| launch.day | 10.910 ± 0.987 | 11.122 ± 0.533 | 11.283 ± 0.176 | 11.256 ± 0.345 |
| launch.month | 3.351 ± 0.025 | 3.370 ± 0.014 | 3.410 ± 0.114 | 3.358 ± 0.025 |
| launch.year | 1.252 ± 0.057 | 1.396 ± 0.032 | 1.268 ± 0.039 | 1.443 ± 0.064 |
| oem | 0.181 ± 0.005 | 0.186 ± 0.003 | 0.178 ± 0.005 | 0.182 ± 0.003 |
| network-edge | 0.230 ± 0.005 | 0.237 ± 0.007 | 0.224 ± 0.008 | 0.232 ± 0.009 |
| model | 0.881 ± 0.004 | 0.862 ± 0.004 | 0.880 ± 0.004 | 0.861 ± 0.002 |

Table 8: Prediction performance on GSM ablated over the mask rate during training. Properties are always masked out randomly, but the probability can be chosen. "Random" means a uniformly random mask rate, newly drawn for each batch. Note that the rate applies only in training.

## C    ARCHITECTURE AND TRAINING DETAILS

Encoders and decoders in the model largely have the same structure, which relies on residual blocks made of a standard $4\times$ hidden layer, a GLU activation, and a post-activation LayerNorm. Parameters are initialized following the Maximal Update Parameterization (Yang et al., 2022). Categorical decoders have the same number of outputs as classes. Numerical decoders, on the other hand, predict GMM parameters, which add up to a total of $3 \times$ Num. Mixtures which is usually a hyperparameter we tune on a validation set. We use the default implementation of the transformer encoder in PyTorch for the entity encoder and the text encoder. Simularly, we use the default transformer decoder for the text decoder. For the text encoder, we use the last layer outputs at the first token as the encoding of the text property. Code for the model architecture, as well as experiments, is available at [github REDACTED].

Preprocessing (for GSMArena and AME2020) includes min-max rescaling for numerical and one-hot encoding for categorical properties. However, we did experiment with semantically encoding the labels of categorical properties using the same tokenization and embeddings from the language modeling component. This yielded interesting results with "semantically meaningful" errors. For instance, if the model never sees a label in the training data, it often predicts a label with a large string intersection with the truth labels. We also experimented with both DICE and trainable periodic embeddings but found no significant difference. Results are reported using periodic embeddings.

We use a cosine annealing schedule for all of our runs. All runs were performed on a handful of V100 GPUs.

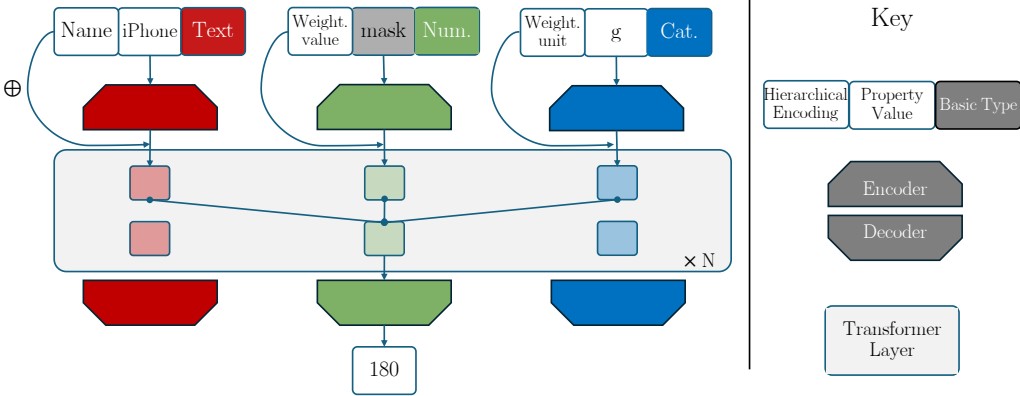

Figure 8: KBFormer architecture on the left and key for the diagram on the right. In this example, the model is tasked to predict the masked value for the `weight.value` property.

The hierarchical encodings are generated by traversing the hierarchy to reach the node for which we would like to compute the prediction (see Figure 9). In our case, we use an RNN to process the sequence and read off the encoding from the hidden representation of the last element in the sequence.

## D    TABULAR DATASETS DETAILS

**Experimental setup**   We use the same experimental setup as Kotelnikov et al. (2023), including the preprocessing and the CatBoost hyperparameters tuned on the validation set of each dataset. For KBFormer, we run a hyperparameter search on learning rate, width, depth, and number of GMM parameters over 100 iterations to optimize the CatBoost performance on the validation set. Finally, we evaluate the test set from the real data after training 10 CatBoost models on 5 realizations of data generated by KBFormer, totaling 50 runs. Table 3 reports the mean and the standard deviation. For the other methods, we re-use the hyperparameters reported by Kotelnikov et al. (2023) and found by tuning each model in a similar fashion.

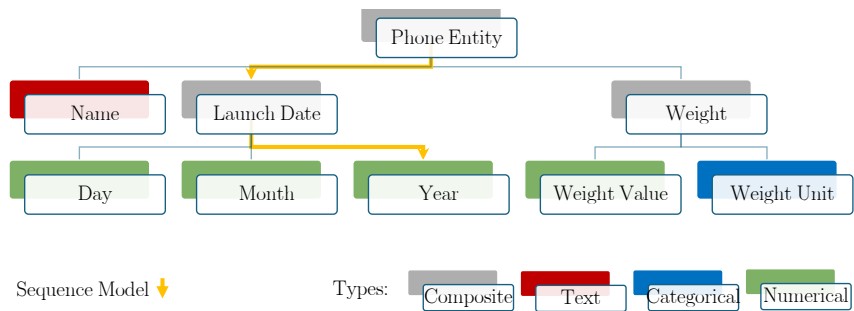

Figure 9: Example phone entity with leaf nodes that have one of the following basic types numerical, categorical, or text. Composite properties are made of other composite types or leaf nodes. Positional encodings are generated at each node using a sequence model over the path that connects it to the root. Encoded values at each node are attended to using the entity encoder

**Baselines** Our main baseline here is TabDDPM, which is also a diffusion model, though it uses very different assumptions (for instance, a uniform kernel instead of one with an absorbing state). These discrepancies and the differences in architecture enable our approach to be better suited to handling numerical quantities, hierarchical and sparse data, and missing values. We also compare against other generative models: TVAE, a tabular variation auto-encoder-based generative model, CTAB-GAN and its upgrade CTAB-GAN+ based on a generative adversarial backbone. Finally, SMOTE is an interpolation method originally proposed for minority oversampling but is used in Kotelnikov et al. (2023) as a sanity-check baseline.

**Datasets** Here is the complete list of the datasets used in this experiment.

| Code | Name | Train size | Val. size | Test size | Num. feat. | Cat. feat. | Task |
|------|------|-----------|-----------|-----------|------------|------------|------|
| ABAL | Abalone | 2672 | 669 | 836 | 7 | 1 | Regression |
| ADUL | Adult | 26048 | 6513 | 16281 | 6 | 8 | Binclass |
| BUDD | Buddy | 12053 | 3014 | 3767 | 4 | 5 | Multiclass |
| CALI | California Housing | 13209 | 3303 | 4128 | 8 | 0 | Regression |
| CARD | Cardio | 44800 | 11200 | 14000 | 5 | 6 | Binclass |
| CHUR | Churn Modelling | 6400 | 1600 | 2000 | 7 | 4 | Binclass |
| DIAB | Diabetes | 491 | 123 | 154 | 8 | 0 | Binclass |
| FB-C | Facebook Comments Volume | 157638 | 19722 | 19720 | 50 | 1 | Regression |
| GEST | Gesture Phase | 6318 | 1580 | 1975 | 32 | 0 | Multiclass |
| HIGG | Higgs Small | 62751 | 15688 | 19610 | 28 | 0 | Binclass |
| HOUS | House 16H | 14581 | 3646 | 4557 | 16 | 0 | Regression |
| INSU | Insurance | 856 | 214 | 268 | 3 | 3 | Regression |
| KING | King | 13832 | 3458 | 4323 | 17 | 3 | Regression |
| MINI | MiniBooNE | 83240 | 20811 | 26013 | 50 | 0 | Binclass |
| WILT | Wilt | 3096 | 775 | 968 | 5 | 0 | Binclass |

Table 9: Dataset description

# E DETAILS ON NUCLEAR DATA

The data is gathered from a live chart of nuclide properties in `https://nds.iaea.org/relnsd/vcharthtml/VChartHTML.html` that is constantly updated. Our snapshot includes all data up to August 2023. Sources for the data are listed in `https://nds.iaea.org/relnsd/vcharthtml/guide.html`, subsection *Sources*. This dataset contains numerical and categorical properties. Numerical properties comprise binding energy, charge radius, the logarithm of the half-life, Spin configuration, abundance of the nucleus in nature, energies available for $\alpha$, $\beta$, $\beta + n$ decays and electron capture (EC), various form factors. The categorical properties are the stability of the nucleus and its parity. We exclude proton/neutron separation energy to prevent binding energy data leakage.

With consistent results across hyperparameter configurations, our chosen model for this task trains for 50,000 epochs with a 0.001 learning rate, no weight decay, 0.1 dropout, and 1024 batch size. It has two encoder/decoder layers per property and 50 GMM components per numerical feature.

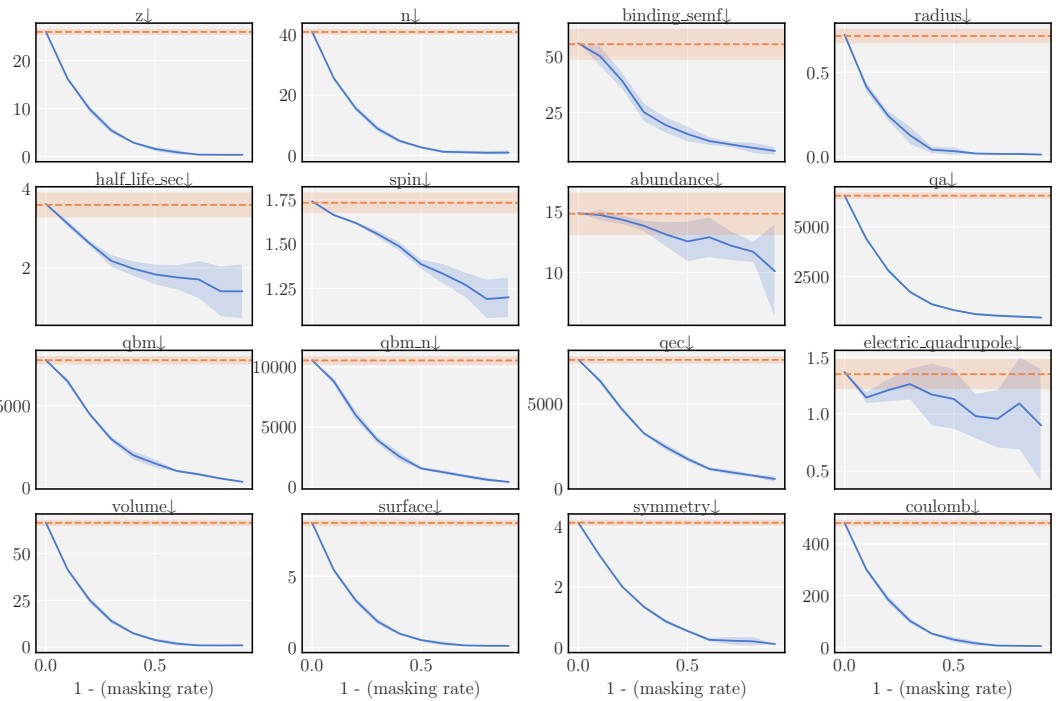

Figure 10: Model performance on a held-out set of the nuclear dataset as a function of masking rate, measured by root mean square error (RMS, ↓) for numerical properties and accuracy (↑) for categorical properties. The dashed baseline reflects always predicting the marginal mode/mean. Error bars are one $\sigma$.

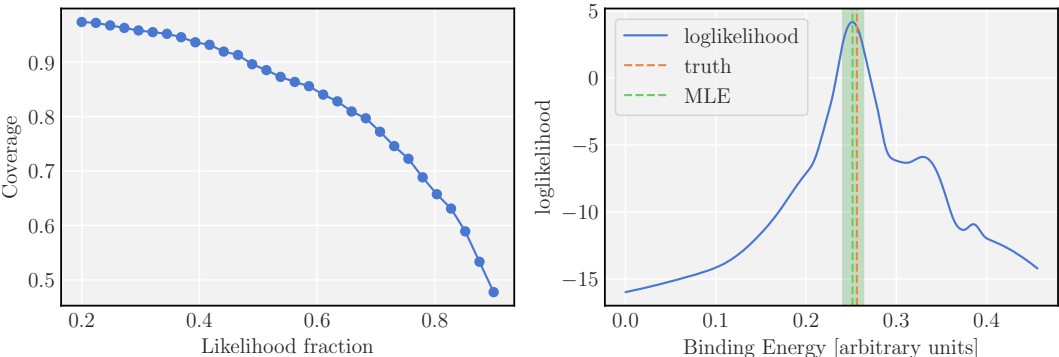

Figure 11: (Left) Coverage of estimated binding energy from the nuclear dataset as a function of maximum loglikelihood fraction contained within the interval. (Right) Full loglikelihood of the binding energy of a validation sample along with the Maximum Likelihood Estimate (MLE) and the $-1/2$ profile likelihood.

## E.1 TRAINING AND EVALUATION ON NUCLEAR DATA

Evaluating the precision of predictions of Tabular Denoising Diffusion Probabilistic Model (TD-DPM) on the nuclear physics dataset requires conditioning on N, Z. Because this cannot be done directly, we generate samples from the joint distribution which includes N and Z and post-hoc condition on samples that are are close to the desired $N$ and $Z$ values (within 0.1 tolerance). We then take the mean prediction of these samples and use it as a model prediction. We used the standard architecture from Kotelnikov et al. (2023) with slightly different hyperparameters, which were tuned

with a validation set on a coarse grid.

As for the GBDT, we handle missing data by filling with the Optimal Constant solution i.e., if a (numerical) categorical property is missing in a particluar sample we simply replace it with the (mean) mode of that property across the training set.

We chose the following hyperparameters for tuning by suggesting suitable values in each distribution:

1. **learning rate**: The learning rate determines the step size taken by the optimizer during training. We used a log-uniform distribution between 0.001 and 1.0.

2. **depth**: The depth of the decision trees in the model. We chose an integer value between 3 and 10 for this parameter.

3. **l2 leaf reg**: The L2 regularization term applied to the objective function. We used a uniform distribution between 0.1 and 10.0 for this parameter.

4. **bagging temperature**: The parameter controlling the intensity of the sampling process for bagging during training. We used a uniform distribution between 0.0 and 1.0 for this parameter.

5. **leaf estimation iterations**: The number of Newton-Raphson iterations for calculating leaf weights. We chose an integer value between 1 and 10 for this parameter.

Additionally, we set some default values for other parameters:

- `iterations`: 2000
- `early stopping rounds`: 50
- `od pval`: 0.001

## F    DETAILS ON THE GSM DATASET

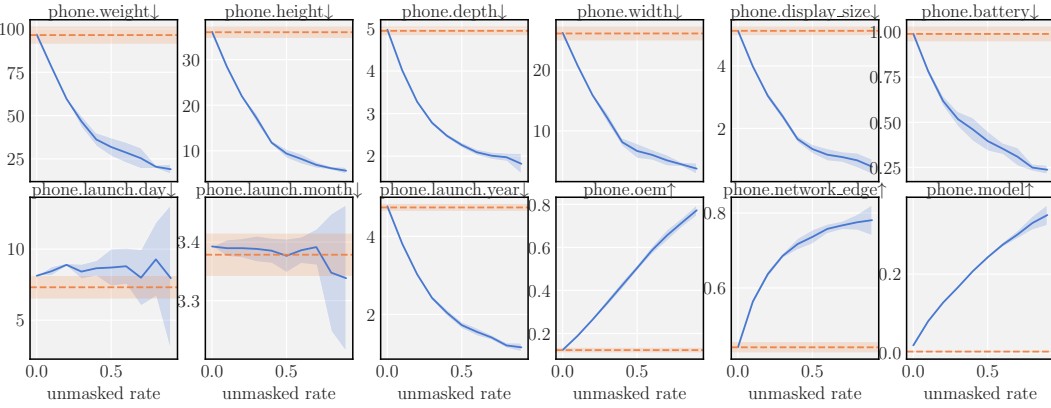

Figure 12: KBFormer performance on a held-out dataset as a function of unmasking rate, measured by root mean square error (RMS, ↓) for numerical properties and accuracy (↑) for categorical properties. The dashed baseline reflects always predicting the marginal mode/mean. Error bars are one $\sigma$.

The GSMArena dataset on phones comes from https://www.kaggle.com/datasets/msainani/gsmarena-mobile-devices, comprises 10679 entries of phone entities, with the following features: model name, OEM name, network edge, weight, display size, height, width, depth, battery and launch date, which is a composite type of day, month and year. The data is split into 80% train and 20% test data. A small number of entries shares both the manufacturer and model name. In such cases, we move the duplicates from the testing set to the training set. This results in a split of about 83% and 17%. The phone launch day entry is fairly sparse, only 5% are filled, but all other values have at leat 85% coverage.

### F.1 GSM TRAINING FOR KBFORMER AND DECODER-ONLY

The custom tokenization for the from-scratch trained decoder defines each possible element in categorical fields as one token. Numerical values are represented as floats with two decimals and tokenization is done per digit. The string representation has special tokens for separation between items, key value separation and separation of hierarchical key. For example, a key like `phone.launch.day` is tokenized as "$T$(phone), $T$(.), $T$(launch), $T$(.), $T$(day)", $T$ representing the token to integer mapping. The model is a 4-layer decoder-only transformer a model dim of 768, 2 heads per self-attention. It is trained with a batch size of 512, a learning rate of 0.0001, weight decay of 0.0001 and no dropout. Those parameters were optimized by sweeping over a coarse grid.

The KBFormer has one encoder and one decoder module with 2 layers each for every feature of the data, a model dimension of 256, 2 heads in each attention, a 2-layer entity encoder and 50 GMM components per feature. It was trained with a batch size of 1024, learning rate of 0.001, no weight decay and a dropout of 0.1 over 20000 epochs. Those parameters were optimized in a similar way as in the decoder procedure.

The Llama model was fine-tuned with LoRA Hu et al. (2022) and FSDP Zhao et al. (2023) via the `llama-recipes` repository (`https://github.com/facebookresearch/llama-recipes`) from Meta AI. Training runs for two epochs, after which the validation loss saturates.

### F.2 GSM TRAINING FOR GBDTS

GBDTs offer state-of-the-art performance on tabular data but they do not handle missing data naturally. To solve this issue, we fill in missing properties with their optimal constant solutions (mean for continuous and mode for discrete). Furthermore, text properties are omitted because GBDTs cannot handle them in a natural fashion. We tune the GBDT hyperparamters on a validation set in a similar way to that of the nuclear physics dataset in Appendix E.1.

## G  LIMITATIONS

While our study demonstrates the efficacy of the KBFormer model in structured generative modeling and high-precision handling of numerical types along with various data types, several limitations and future research directions emerge:

1. **Scalability and Pre-training Challenges:**
   - *Current Scope:* The model's current application is confined to datasets of a limited scale.
   - *Future Aspirations:* Aiming to scale the model to joint training on larger and more varied datasets introduces significant challenges, especially in pre-training.

2. **Integration with Language Models:**
   - *Current Integration:* The model has a strong capacity in handling structured data and generating high-precision predictions but uses a small transformer to model text properties.
   - *Future Potential:* Extending our approach to integrate with LLMs could enhance performance on knowledge-intensive tasks and benchmarks.

3. **Generalization within Knowledge Graphs:**
   - *Current Methodology:* The model treats static entities as independent units, not fully leveraging relational dynamics within a knowledge graph.
   - *Future Exploration:* Investigating how the model can achieve generalization within the context of a knowledge graph is critical.

4. **Knowledge Representation in Foundation Models:**

- *Broader Implications:* Current foundation models, including LLMs, store knowledge in latent forms that are not human-interpretable or easily editable.
- *Future Directions:* Developing structured knowledge models to augment LLMs, aiming for explicit, interpretable, and editable knowledge representation, remains a pivotal challenge.

