# OpenReview forum: "KBFormer: A Transformer-based Diffusion Model of Structured Entities with Heterogeneous Properties"
_ICLR.cc/2024/Conference — Submitted to ICLR 2024_

### Official Review · Reviewer_b6La · 2023-10-31

**Soundness:** 3 good
**Presentation:** 2 fair
**Contribution:** 2 fair
**Rating:** 6
**Confidence:** 4

**Summary:**

The paper proposes a generative attention-based architecture, named KBFormer, for modeling structured entities consisting of different property types. The model is able to perform joint modeling of an entity's properties due to the hybrid diffusion training approach. The experimental results on a device KB and a nuclear physical dataset demonstrate the model's capability in representation learning for entity completion in diverse settings. The model is demonstrated to share information across properties and have various downstream uses for scientific applications. The inherent probabilistic nature of the model enables predictions accompanied by uncertainties.

**Strengths:**

Originality: The paper uniquely contributes to the field by proposing a generative attention-based architecture which can handle heterogenous data types through a mixed continuous-discrete diffusion process over the properties. Such a method stands out among the popular autoregressive models.

Quality: The paper carefully designs the experiments to support the idea and improve their illustrations with helpful visualizations.

Clarity: The paper effectively communicates its ideas and findings with clarity. The paper is well-written, and the logic is coherent. Necessary mathematical deductions are presented for better understanding of the diffusion process.

Significance: The model proposed in the paper successfully handles heterogeneous property types along with hierarchical encodings with semantic meaning for different property types, and demonstrates its potential in various downstream scientific tasks. Besides, the model has an edge over traditional autoregressive models due to its inherent probabilistic nature.

**Weaknesses:**

1. Although the authors make clear introductions to the hybrid diffusion training paradigm, the explanation for the model architecture is not clear enough (sometimes even confusing). In fact, the first of my two questions is because of not understanding the architecture here. I suggest the authors can modify Figure 5 and 6 to make the pipeline transparent to readers and include more details in the texts in an ordered way.

2. Although the inherent probabilistic nature of the model makes it suitable for prediction tasks for scientific scenario, I still believe that comparisons between the KBFormer and some regression model baselines should be included to demonstrate the effectiveness of the model. Besides, in order to illustrate the model's capability in scientific applications, more experiments on benchmarks from other disciplines like molecules, proteins, etc., are required.

**Questions:**

1. In terms of the model architecture, I suppose the output from the encoder is $(3, feature size)$ for the example shown in Figure 5. If that is true, I'm wondering why it is necessary to use a transformer model with such a short "sequence length".

2. Transformer encoders are used to encode text fields. However, if the text is simply a property value (e.g. "iPhone" in Figure 5), why not just use a pretrained word embedding?

3. The diffusion training strategy is a promising solution for probabilistic-based generative models. However, is the diffusion paradigm the optimal method to do this? Is there any other generative model (or even regressive model) that does not rely on the noise and denoise process?

---

> ### Author Response · Authors · 2023-11-16
>
> Thank you for the thorough review and suggestions!
>
> > 1. Although the authors make clear introductions to the hybrid diffusion training paradigm, the explanation for the model architecture is not clear enough (sometimes even confusing). In fact, the first of my two questions is because of not understanding the architecture here. I suggest the authors can modify Figure 5 and 6…
>
> We have made significant strides in improving the clarity of the text and the model architecture in particular (see changes in blue). Following your suggestions, we completely reworked Figures 5 and 6 (now combined into Figure 4).
>
> > 2. Although the inherent probabilistic nature of the model makes it suitable for prediction tasks for scientific scenario, I still believe that comparisons between the KBFormer and some regression model baselines should be included to demonstrate the effectiveness of the model. Besides, in order to illustrate the model's capability in scientific applications, more experiments on benchmarks from other disciplines like molecules, proteins, etc., are required.
>
> Thank you for the suggestions! We have now included an additional baseline: Gradient Boosted Decision Trees (GBDT)s, which offer state-of-the-art performance on tabular data for both regression and classification tasks. Our experiments show that we can perform competitively against specialized regression models even though our model is generative in nature. We also included additional experiments across 15 datasets and compared our results against other generative models spanning various tasks, including two scientific datasets (MiniBooNE and Higgs).
>
> > Q1. In terms of the model architecture, … why it is necessary to use a transformer model with such a short "sequence length".
>
> We have updated this figure and added extensive details around it. The previous figure used a small example with three properties for simplicity, though in practice, we use many more (see Table 6 with a description of all the datasets we used and the number of properties in each one.) The updated figure should clarify that there are more properties than what is shown. The transformer aggregates information across all properties (e.g., our tabular data experiments have up to 50). In future work, we aim to scale to a much larger number of properties and knowledge bases, and the transformer architecture is a good candidate for this.
>
> > Q2. Transformer encoders are used to encode text fields. However, if the text is simply a property value (e.g. "iPhone" in Figure 5), why not just use a pretrained word embedding?
>
> Figure 5 (now 4) should no longer imply that text fields are single words. If this were the case, we could indeed use pre-trained word embeddings. However, the goal of the text generator is to fill text property values with a (possibly long) sequence of tokens.
>
> > Q3. The diffusion training strategy is a promising solution for probabilistic-based generative models. However, is the diffusion paradigm the optimal method to do this? Is there any other generative model (or even regressive model) that does not rely on the noise and denoise process?
>
> In our latest round of experiments, we explored the performance of our diffusion-based approach against several other generative models including VAEs, GANs, and a different implementation of diffusion on tabular data (TabDDPM). We also compare against regression models for nuclear physics and the GSM phone dataset. Across all the experiments, KBFormer obtains favorable performance, which illustrates the strength of the approach. Please note that our diffusion process has “less noise” than expected from more standard approaches. When we make predictions autoregressively, noise is injected via the random order in which properties are unmasked.

---

> > ### Comment · Reviewer_b6La · 2023-11-19
> >
> > Thanks for the author's response to my questions and concerns. I really appreciate that you have helped me better understand your contributions. The responses have resolved my concerns, and the quality of the paper is significantly improved. I'll consider raising my score.

---

### Official Review · Reviewer_QYBP · 2023-11-01

**Soundness:** 2 fair
**Presentation:** 3 good
**Contribution:** 2 fair
**Rating:** 5
**Confidence:** 2

**Summary:**

This paper proposes a generative attention-based architecture that models structured entities comprising different property types, with applications on KB entities and tabular data. A hybrid diffusion training paradigm is proposed to handle the modeling of heterogeneous properties.

**Strengths:**

1. The paper is well-presented, with excellent visualizations and clear delivery of the model details and results.
2. KBTransformer enjoys superior performance against baselines in terms of prediction accuracies on two real data sets.

**Weaknesses:**

1. The paper's contribution seems to be a bit incremental, since the diffusion modeling over heterogeneous data in Section 3.2 follows the previous work [1]. It would be helpful if the authors could clarify the difference/contribution of the proposed method.
2. In the experiments, only the baseline that always predicts the marginal mode/mean is compared in terms of the prediction accuracy with different unmasked rates. Are there any other baselines with regression/autoregression that can be compared? Also it would be helpful if the authors could elaborate on Section 3.1 about why other traditional models with regression and masked modeling will have less optimal performance.

---
[1] Jacob Austin, Daniel D Johnson, Jonathan Ho, Daniel Tarlow, and Rianne Van Den Berg. Structured denoising diffusion models in discrete state-spaces. Advances in Neural Information Processing Systems, 34:17981–17993, 2021.

**Questions:**

Please see above.

---

> ### Author Response · Authors · 2023-11-18
>
> Thank you for your review! We uploaded a revised paper with additional clarifications, baselines, and experiments, which we hope address your concerns.
>
> > 1. The paper's contribution seems to be a bit incremental, since the diffusion modeling over heterogeneous data in Section 3.2 follows the previous work [1]. It would be helpful if the authors could clarify the difference/contribution of the proposed method.
>
> Here we summarize the differences between our work and [1]:
>
> D3PM [1] deals with discrete-state spaces in **discrete time.** Campbell’s formulation [[2]](https://proceedings.neurips.cc/paper_files/paper/2022/file/b5b528767aa35f5b1a60fe0aaeca0563-Paper-Conference.pdf) (which itself is parallel to [1]) gives a very general case for **continuous-time** discrete-state models. Our derivation builds on the continuous-time approach and shows that for the absorbing-state case, the ELBO can be estimated using a single evaluation of the model (two are required in Campbell’s formulation, or a biased approximation). Additionally, [1] and [2] deal with continuous state spaces via Gaussian discretization, whereas we propose using a Gaussian Mixture Model.
>
> Moreover, in addition to the diffusion paradigm we derive, we also propose an architecture and hierarchical encoding scheme that can handle structured data and obtain superior performance across most datasets.  Finally, note that in many of our new experiments, we compare against TabDDPM, which is partly an implementation of [1] to handle tabular datasets, and show that we obtain favorable performance in many cases.
>
> > In the experiments, only the baseline that always predicts the marginal mode/mean is compared in terms of the prediction accuracy with different unmasked rates. Are there any other baselines with regression/autoregression that can be compared? Also it would be helpful if the authors could elaborate on Section 3.1 about why other traditional models with regression and masked modeling will have less optimal performance.
> >
>
> Yes! We added Gradient Boosted Decision Trees (GBDTs) as a new baseline in our reconstruction experiments. We train a GBDT on all other properties for each numerical and categorical type. This is a much stronger baseline than what we had before (see the updated section 5 and appendices E and F, which contain more training details). We also include experiments on 15 datasets with new baselines built on various generative frameworks (GAN, VAEs, DDPM, etc).
>
> As for section 3.1, we argued that generating many properties at once can degrade performance. In our diffusion framework, that would be akin to skipping “time steps.” We ran ablations on the 15 datasets to show that generating all properties at once can impact performance greatly (see Table 4 in Appendix B.1). We also show a few qualitative examples in Appendix B.1.2). In one of these examples, we generate MNIST digits in one time step, and one can visually inspect the samples to see they are a lot noisier compared to the autoregressive samples. To clarify further, suppose in a language modeling setting, we are interested in predicting a sequence of two tokens that answers a question positively. “[Of] [course]” and “[No] [problem]” could be equally likely candidates. However, sampling the two tokens simultaneously can result in answers like “[No] [course]” or “[Of] [problem].” We run exactly into this inconsistency issue if we try to sample all properties simultaneously. But it can be solved by conditioning the second token generated on the first token. This is why our autoregressive model performs much better than the single-step approach.
>
> [1] Jacob Austin, Daniel D Johnson, Jonathan Ho, Daniel Tarlow, and Rianne Van Den Berg. Structured denoising diffusion models in discrete state-spaces. Advances in Neural Information Processing Systems, 34:17981–17993, 2021.
>
> [2] Andrew Campbell, Joe Benton, Valentin De Bortoli, Thomas Rainforth, George Deligiannidis, and Arnaud Doucet. A continuous time framework for discrete denoising models. Advances in Neural Information Processing Systems, 35, 2022.

---

### Official Review · Reviewer_yZ6H · 2023-11-06

**Soundness:** 3 good
**Presentation:** 3 good
**Contribution:** 4 excellent
**Rating:** 6
**Confidence:** 3

**Summary:**

The paper presents KBFormer, an innovative transformer-based diffusion model adept at managing structured entities featuring heterogeneous characteristics. This versatile model excels in accommodating entities with complex hierarchical attributes, making it particularly suited for structured knowledge bases (KBs) and tabular data. KBFormer is designed to learn representations that are effective for entity completion across a range of contexts, and its probabilistic approach allows for predictions that incorporate measures of uncertainty. The authors detail the model's training methodology, introducing a novel loss modification that reimagines the problem as a continuous-time diffusion process over discrete states with an absorbing state. The paper culminates with an exploration of KBFormer's applicability in downstream tasks, highlighting its capacity to model numerical properties with remarkable precision and to generate accurate predictions in specialized domains, including nuclear physics.

**Strengths:**

- This work introduces KBFormer, a novel transformer-based diffusion model adept at managing structured entities with varied and complex properties.
- The paper demonstrates the model's practical applications, particularly in high-accuracy numerical property modeling and precise prediction-making in fields like nuclear physics, highlighting its value to researchers.
- The paper is well-placed in the literature, with the KBFormer framework being noted for its flexibility and extensiveness, marking a progression from previous models.
- The paper is clearly written and accessible, with lucid explanations of the model's architecture, training processes, and experimental results, catering to a broad audience.
- The submission includes supplementary materials for implementation, which enhances the paper's credibility and supports the reproducibility of the KBFormer model.

**Weaknesses:**

- This paper lacks ablation studies. The paper does not include ablation studies to analyze the contribution of different components of the model to its overall performance. For example, it is mentioned in paragraph "Encoding" of section 4, that two alternatives for embedding numerical values, yet it lacks a quantitative performance comparison between these methods. Conducting such an analysis could shed light on the critical components of the model and direct future enhancements.
- There is no discussion in the paper about the limitations or potential failure modes of the KBFormer method. Including this could be crucial for fully understanding where the model may fall short in practical applications and where further research and development could be most beneficial.

**Questions:**

Besides the aspects mentioned in "Weakness", I have the following concerns:

- I would like to suggest the authors confirm whether the abbreviations used throughout the paper, such as “KB” in the abstract, are consistently defined upon first use? A uniform approach to abbreviation would aid reader comprehension.
- The introduction promises that the KBFormer model addresses several tasks: KB completion, entity linking, anomalous property detection, and enhancement of foundation models with learned representations. The experiments, however, seem to showcase a subset of these. Can the authors clarify the criteria for experiment selection and indicate if demonstrating the model's capabilities on the remaining tasks is within the scope of future work?
- Would the authors consider revising the reference list to ensure that all citations are consistent and reflect the most current research where applicable? This would help maintain the paper’s relevance and assist readers in locating the sources.

---

> ### Author Response · Authors · 2023-11-16
>
> Thank you for your review and all the suggestions! We are currently working on implementing them. We would be happy to revise our references with additional material. Are there any specific references we are missing?

---

> > ### Comment · Reviewer_yZ6H · 2023-11-16
> >
> > Dear Authors,
> >
> > Thank you for addressing my query. While I am satisfied with the overall selection of relevant literature, I would like to recommend a minor yet important refinement in the consistency of the reference list. Specifically, I've noticed some discrepancies in how papers from the 'Advances in Neural Information Processing Systems' are cited. For instance, some entries include page numbers, while others do not, and there are a few that even have links to the papers. Uniformity in these citations would greatly enhance the clarity and professionalism of the reference list. This suggestion, although seemingly minor, reflects a commitment to meticulous academic standards. (Yeah well that might be my OCD.)
> >
> > Warm regards,
> > Reviewer yZ6H

---

> ### Author Response · Authors · 2023-11-18
>
> Thank you for your review and the great suggestions. The new version of the paper reflects the changes mentioned below.
>
> > • This paper lacks ablation studies. The paper does not include ablation studies to analyze the contribution of different components of the model to its overall performance. For example, it is mentioned in paragraph "Encoding" of section 4, that two alternatives for embedding numerical values, yet it lacks a quantitative performance comparison between these methods. Conducting such an analysis could shed light on the critical components of the model and direct future enhancements.
>
> You are absolutely right! We have added ablations in Appendix B for the following:
>
> - Single-step masked modeling vs. absorbing state (autoregressive) diffusion.
> - DICE vs. Periodic Encoding for the numerical embeddings.
> - GMM vs. MSE for numerical quantities. (Using GMM with 1 mixture and unit variance is equivalent to training with MSE.)
> - Tying numerical embeddings across properties.
>
> > • There is no discussion in the paper about the limitations or potential failure modes of the KBFormer method. Including this could be crucial for fully understanding where the model may fall short in practical applications and where further research and development could be most beneficial.
>
> When mentioning future work, we allude to scaling to larger datasets and performing pre-training across multiple domains. That is our current main limitation. We made it more explicit in the paper and added a longer limitations section in Appendix G.
>
> > • I would like to suggest the authors confirm whether the abbreviations used throughout the paper, such as “KB” in the abstract, are consistently defined upon first use? A uniform approach to abbreviation would aid reader comprehension.
>
>
> This should be resolved in our latest update of the paper. Apologies for missing some definitions in the first version!
>
> > • The introduction promises that the KBFormer model addresses several tasks: KB completion, entity linking, anomalous property detection, and enhancement of foundation models with learned representations. The experiments, however, seem to showcase a subset of these. Can the authors clarify the criteria for experiment selection and indicate if demonstrating the model's capabilities on the remaining tasks is within the scope of future work?
>
> In the introduction, we list some applications to motivate developing our approach, though it is true we do not tackle all tasks we use for motivation. However, we are working on extensions to some of the other tasks in future work.
>
> > • Would the authors consider revising the reference list to ensure that all citations are consistent and reflect the most current research where applicable? This would help maintain the paper’s relevance and assist readers in locating the sources.
>
> We updated our references for improved consistency across references from the same source.
>
> Cheers! The authors

---

> > ### Comment · Reviewer_yZ6H · 2023-11-20
> >
> > Dear Authors,
> >
> > Thank you for addressing the concerns raised by me. The updates to the paper with so many details on ablation studies are quite impressive, leading me to consider raising my review score. I will pay close attention to other reviewers' responses. I appreciate your responsiveness to the feedback.
> >
> > Best regards,
> > Reviewer yZ6H

---

### Official Review · Reviewer_vWit · 2023-11-08

**Soundness:** 2 fair
**Presentation:** 2 fair
**Contribution:** 3 good
**Rating:** 5
**Confidence:** 2

**Summary:**

This paper proposes a diffusion-based generative model to study the structured entities, with various property types such as numerical, categorical, strings. This work should address an interesting topic. However, I am not an expert in this area, and I am getting quite confused about the whole procedure and mechanism after several rounds of reading. I have a quick look at the code and believe the author should implement correctly. Maybe the authors can provide more details about KBformer, and re-organize the presentation to make it easy for understanding.

**Strengths:**

**1** The problems that this work tries to address are interesting and important. Also, the performance is promising.

**1** The usage of diffusion process for entities is well-explored.

**Weaknesses:**

However, I hope the authors can improve the paper through the following aspects:

**1** the presentation of the paper is quite poor. For example, I hope the authors can make it comprehensive for Figure 5, many parts in this figure is unexplained. Maybe since I am not an expert in this topic, I find it quite confused and have many unclear parts for the KBFomer architecture.

**Questions:**

NA

---

> ### Author Response · Authors · 2023-11-16
>
> Thank you for your review! We elaborated extensively in the paper (see changes in blue). We hope you find the updated Figure 5 (now Figure 4) to be clearer. If not, please let us know what points remain unclear so we can clarify them both in the paper and our final response.

---

### Official Review · Reviewer_EEBS · 2023-11-08

**Soundness:** 3 good
**Presentation:** 2 fair
**Contribution:** 3 good
**Rating:** 6
**Confidence:** 3

**Summary:**

This work proposes a diffusion model for generating and demasking structured entities. KBFormer uses a mixed continuous-discrete diffusion process to generate different data types, such as text, numerical, categorical, and ordinal, and a transformer module to model cross-property relations. The paper demonstrates that KBFormer can outperform large language models on entity completion tasks and provide uncertainty estimation.

**Strengths:**

- The proposed architecture with much smaller model parameters  can outperform LLMs which highlights the importance of modeling structure-aware inductive biases.

- It can perform entity completion tasks with high accuracy and provides uncertainty estimation which is very useful for science applications that require confidence and reliability

- It serves as an interpretable multi-modal foundation model for structured data and can augment LLMs with structure-aware inductive biases.

**Weaknesses:**

1) The paper does not discuss or compare with other methods that can handle discrete data with continuous state, such as [1].  Moreover, the paper only compares with LLaMA2 for the first experiment, but it would be interesting to see how the proposed model performs against other knowledge masking strategies, such as [2, 3].

2) The section on “Continuous Relaxation of Discrete State Diffusion” is not well explained. It is unclear what its objective is;  is the goal to learn bin centers, and how they are used in demasking? Is the discretization with 256 bins and learned bin centers similar to GMM with 256 mixtures? The paper also introduces some terms without proper definitions, such as “an infinite bin limit approximation” and “discretization with a large but finite bin density”. It would be helpful to provide more details and intuition behind these concepts.

3) The paper needs to improve its writing quality and clarity. Some specific issues are:
Proposition 1: The font of the proposition should be consistent and italicized. Is a proof provided  in the appendix?
Page 6: The phrase “… see Section 3.2.” should be enclosed in parentheses, as it is not part of the main sentence.

References:

[1] Chen, T., Zhang, R. and Hinton, G., 2022. Analog bits: Generating discrete data using diffusion models with self-conditioning. arXiv preprint arXiv:2208.04202.

[2] Sun, Yu, et al. "Ernie: Enhanced representation through knowledge integration." arXiv preprint arXiv:1904.09223 (2019).

[3] Wang, Ruize, et al. "K-adapter: Infusing knowledge into pre-trained models with adapters." arXiv preprint arXiv:2002.01808 (2020).

**Questions:**

1) Which type of encoder and decoder did you use for different property types: i)- conditioning on the property itself or ii) disjoint encoders for each property ?

2) In page 6, it is stated that “ ‘year’ has the same representation in different contexts …” but the RNN encoder outputs context-aware representation. Can you clarify?

3) For experiments in section 5.2, what is the evaluation method? It is also worth comparing it with other models such as fine-tuned LLaMA2.  Also, for the evaluation of experiment in 5.1, please provide more clarification or examples for rotating each dictionary’s fields D times and predicting only the last value.

---

> ### Author Response · Authors · 2023-11-16
>
> We appreciate your thorough feedback!
>
> > 1. The paper does not discuss or compare with other methods that can handle discrete …
>
> Thank you for bringing this literature to our attention. In this submission, we focus on handling different property types in a generative framework; in future work, we will focus on knowledge-intensive tasks, making these comparisons very important.
>
> In the latest submission, we conducted additional experiments across 15 datasets, comparing KBFormer against several other generative models specializing in structured data. The additional results showing favorable performance on most datasets against CTABGAN, TVAE, and TabDDPM are presented in Table 3. We hope that these new experiments satisfy your request for more baselines.
>
> > 2. The section on “Continuous Relaxation of Discrete State Diffusion” is not well explained…
>
> We have changed the contents of this section and added significantly more details to improve the clarity and the general flow (see the new text in blue).
>
> To summarize here, the objective of this section is to give some intuition on how one might utilize the formulation of discrete space diffusion for continuous properties. Naively, if one wants to model a continuous property as a multi-class prediction problem, one would discretize the space with many bins (to maintain good precision), but the softmax computation can become quite expensive. This would also discard ordinality. However, under the simplifying assumption of Gaussian properties, we can formulate the problem as a mean-squared error (MSE) and maintain ordinality and good precision. This assumption does not usually hold, so we must use something more expressive, like a Gaussian Mixture Model (GMM), which can be seen as a generalization of the MSE approach. This section aims to develop intuition to justify using GMMs to model numerical properties within the full framework.
>
> In practice, we completely drop any discretization and use the GMM parameters to sample numerical properties instead of logits. The number of mixtures we choose is simply a hyperparameter.
>
> Please let us know if anything remains unclear so we can elaborate in further responses and the paper.
>
> > 3. The paper needs to improve its writing quality and clarity. Some specific issues are: Proposition 1: …
>
>
> We expanded our explanations and improved the general flow (see changes in blue). We now clearly point to the proof of Proposition 1 in the appendix, and we fixed all the typos/issues we found.
>
> > Q1. Which type of encoder and decoder did you use for different property types: i)- conditioning on the property itself or ii) disjoint encoders for each property?
>
> One can do both. On a larger scale, we expect conditioning to enable dealing with unseen properties, but for the experiments in this paper, we used disjoint encoders/decoders. We ablate performance when sharing numerical embeddings in the current framing on GSM, which also seems to work.
>
> > Q2. In page 6, it is stated that “ ‘year’ has the same representation in different contexts …
>
> Yes, that was unclear. Apologies for the confusion. The GSM dataset has the property "Phone.Launch.Year". The RNN will process this as a sequence, RNN(["Phone", "Launch", "Year"]). Thus, if the token "Year" exists in another context (let's say "Phone.Discontinued.Year"), the RNN representations will contain information on the similarity of both keys.
>
> > Q3. For experiments in section 5.2, what is the evaluation method? It is also worth…
>
> We added a strong baseline across both sections referenced. In (prior) section 5.2, we evaluated the model's predictions on a given property when conditioned on all other available properties (this dataset is sparse).
>
> In (prior) 5.1, we explained why we rotate the inputs as an augmentation. To answer the question here, since LLaMA is sequence-autoregressive, we have to rotate the string representation of the dictionary such that each field/property can appear in the last position. This ensures we can predict a value for the property while conditioning on all other properties.
>
> More generally, causal models can perform worse when predicting properties [[1]](https://arxiv.org/pdf/2309.14402.pdf), even when they are in the training set. So, we ensure that the model has seen enough permutations of the properties to be able to make predictions while conditioning on any subset.
>
> [1] Physics of Language Models: Part 3.2, Knowledge Manipulation

---

### Author Response · Authors · 2023-11-16

First, we thank the reviewers for their insightful comments and suggestions. We have updated the paper in an effort to engage in further discussion before the rebuttal period ends. The experiments, changes, and additions we refer to in the comments to the reviews can be found in blue text. While we continue working on the reviewers’ suggestions, we hope this enables the reviewers to bring up potential outstanding issues (particularly with clarity) for us to address.

---

### Author Response · Authors · 2023-11-22
**Rebuttal Summary**

We want to thank the reviewers again for their constructive feedback. Based on the suggestions, we believe we have improved the submission substantially. In particular, we supplemented our empirical studies with detailed comparisons against other generative models on 15 tabular datasets, added a strong baseline to the original experiments, improved the general clarity of the paper, and updated our figures.

For convenience, we will summarize the common concerns of some of the reviewers and explain the steps we took to resolve each individually. Changes in the updated submission were highlighted in blue to make them easier to identify.

> Additional baselines and comparison with regression models

1. We provide an additional baseline using multiple Gradient Boosted Decision Trees (GBDTs) trained in an all-but-one setup (only the property that needs to be predicted is left out) and compare it against KBFormer for the nuclear physics task and the GSMArena dataset. This should be a strong baseline because we train a specialized model for each feature and compare it against our generative model conditioned on the same features used for training the GBDT. Both models offer comparable prediction performance where text properties are not involved. KBFormer and the GBDTs perform better than the other generative models trained.
2. We further add results on 15 benchmarks. The task is to generate synthetic data learned from a real dataset. The quality of the synthetic data is evaluated by training a downstream model on the synthetic samples to perform a supervised learning task (regression or classification). The performance of the downstream model (a GBDT in this case) is evaluated on a test set from the original data. We compare against several structured generative models/interpolation techniques (TabDDPM, TVAE, CTAB-GAN, etc.). Here, we improve upon the prior state-of-the-art on most datasets.

    The results from the above experiments have been added to Section 5, and details about the additional datasets and the experiments conducted are reported in Appendix D.


> Writing quality and clarity, specifically in Sections 3 and 4 (ensure the style is consistent and abbreviations are introduced, etc.)

1. Added many details and improved the general flow of the paper.
2. Improved the overall clarity of the architecture and the training procedure in Section 4.
Additional training, evaluation, and dataset details can now be found in Appendices C-F.
3. Updated and improved Figures 5 and 6 (now merged into Figure 4).
4. Minor inconsistencies and missing abbreviations have been resolved, and the references have been revised.
5. Elaborated on Failure modes and areas of further research (see also Appendix G).

> Additional ablation studies

Here, we perform extensive ablations of various components:

1. Single-step masked modeling vs. Autoregressive predictions based on diffusion with an absorbing state kernel.
2. DICE vs. Periodic Embeddings for numerical embeddings.
3. Tied numerical embeddings across properties.
4. GMM vs. MSE to model numerical types.
The conclusions are detailed in Appendix B.


We hope that these additions adequately address the concerns raised.

The Authors

---

### Meta-Review · Area_Chair_NPgt · 2023-12-09

**Metareview:**

The work proposes to use transformers to deal with typed-datasets, and to incorporate the hierarchical information over entities and their properties. To do so, authors propose a continuous relaxation of a discrete diffusion as a generative process. The result, KBFormer, is evaluated in generative and predictive tasks on heterogeneous tabular data.

The reviewers appreciated the direction of KBFormer and the task of having generative models of tabular data that are aware of the heterogeneous and hierarchical structure of data. At the same time they raised concerns about the presentation, the possible incrementality of the contribution and the lack of experimental details and further ablations. Authors greatly revised the paper during the rebuttal, improving the presentation and clarifying certain obscure parts in the methodology (e.g., end of Sec 3 and Sec 4). However, this proved not enough for reviewers to change their opinion. I agree that the aforementioned concerns are still valid, especially the comparison with previous transformers and diffusion processes for heterogeneous data. I encourage authors to take into account the concerns of the reviewers (especially b6La and QYBP).

The paper is therefore rejected.

**Justification For Why Not Higher Score:**

While the presentation improved a lot, many sections are still not detailed enough and leave the reader to wonder why certain model choices have been made.

**Justification For Why Not Lower Score:**

N/A

---

### Decision · Program_Chairs · 2024-01-16

Reject